# Remission spectroscopy resolves the mechanism of action of bedaquiline within living mycobacteria

Suzanna H. Harrison[1,2,6], Rowan C. Walters[1,2,6], Chen-Yi Cheung[3], Roger J. Springett[1,2,4], Gregory M. Cook[3,5], Morwan M. Osman[1,2,6] & James N. Blaza[1,2]

Bedaquiline, an ATP synthase inhibitor, is the spearhead of transformative therapies against drug-resistant *Mycobacterium tuberculosis*. Here, we use remission spectroscopy to measure the energy-transducing cytochromes within unperturbed, respiring suspensions of mycobacterial and human cells, allowing spectroscopic measurements of electron transport chains as they power living cells and respond to bedaquiline. No evidence is found for protonophoric or ionophoric uncoupling. Rather, by directly inhibiting ATP synthase, bedaquiline slows the respiratory supercomplex (Qcr:Cta; *bcc:aa₃*) by increasing the proton-motive force, causing sub-second redirection of electron flux through the cytochrome *bd* oxidase (CydAB) to $O_2$. Electron flux redirection explains the idiosyncratic bedaquiline-induced increase in $O_2$ consumption rates previously observed. Redirection occurs as CydAB is present even in cells grown in plentiful $O_2$. Applying the same approach to human cells did not detect bedaquiline-induced inhibition of mitochondrial function despite such inhibition being seen in isolated systems. Overall, we clarify how bedaquiline works, why different models for its action developed, and the mechanisms underlying the synergy of bedaquiline in combination regimes.

Tuberculosis is the world's worst infectious killer: 1.3 million people die of TB annually, with another 10 million becoming infected[1]. When approved, bedaquiline was the first new drug in over four decades, and as part of the BPaL regime (bedaquiline, pretomanid, linezolid) it has transformed the treatment of drug-resistant TB[2,3]. Resistance to bedaquiline is rapidly growing and there is evidence of bedaquiline-resistant *Mycobacterium tuberculosis* reinfecting new patients[4].

Bedaquiline targets ATP synthase of the mycobacterial bioenergetic system[5,6]. Bedaquiline immobilised on a column binds to mycobacterial ATP synthase[7] and in vitro resistance mutants localise to AtpE (the ring-forming 'c' subunit composed of a circular c₉ ring in the $F_O$ membrane domain)[6]. The binding site has been confirmed in a 1.7 Å crystallographic map of the isolated c-ring from *Mycobacterium phlei*, and in cryo-EM maps of the full enzyme from *Mycobacterium smegmatis* and *M. tuberculosis*[8–10]. Worryingly, bedaquiline inhibits human ATP synthases in vitro, binding in a similar pose[10,11]. When *M. tuberculosis* is exposed to bedaquiline, ATP depletion starts to occur within hours but cell death takes around a week, consistent with metabolic remodelling allowing mycobacteria to survive in a stressed state that is ultimately overwhelmed[12]. Induced metabolic changes include the bypass of ATP-demanding enzymes such as phosphofructokinase, upregulation of the glyoxylate shunt, and secretion of metabolites such as succinate[13].

[1]York Structural Biology Laboratory, Department of Chemistry, University of York, York, UK. [2]York Biomedical Research Institute, University of York, York, UK. [3]Department of Microbiology and Immunology, University of Otago, Dunedin, New Zealand. [4]Cellspex Ltd, Northamptonshire, UK. [5]School of Biomedical Sciences, Queensland University of Technology, Brisbane, QLD, Australia. [6]These authors contributed equally: Suzanna H. Harrison, Rowan C. Walters, Morwan M. Osman. ✉e-mail: morwan.osman@york.ac.uk; jamie.blaza@york.ac.uk

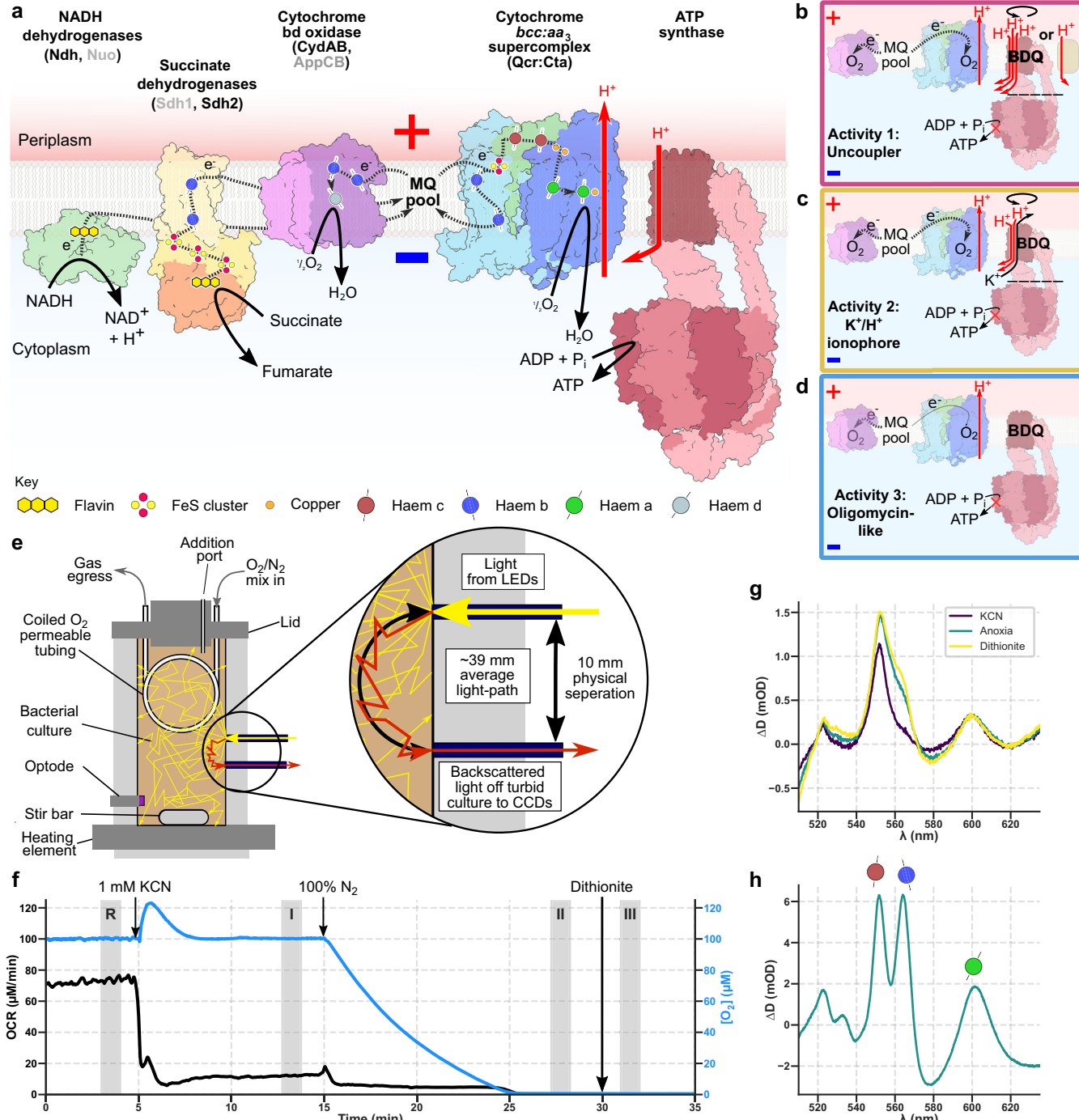

**Fig. 1 | The bioenergetic chamber allows measurements of cytochrome spectra and oxygen consumption in living cells. a** The mycobacterial oxidative phosphorylation system, including abundant cytochromes, NDH-2, and ATP synthase. The cartoon is based on structural and functional information on NADH dehydrogenases[73,74], Sdh2[75,76], the *bd* oxidase (Cyd)[19,20], the respiratory supercomplex[77–79], and ATP synthase[9,10]. Multimeric complexes are shown in their monomeric form. The names of additional complexes that catalyse the same reactions but are less abundant are in grey; AppCB is found in *M. smegmatis* but not *M. tuberculosis*. **b–d** Schematics of the three activities tested in this work. **e** The bioenergetic chamber, showing how the components interface with the bacterial culture. **f** A representative experiment in the bioenergetic chamber. The chamber contained *M. smegmatis* (7H9 media + 50 mM glycerol) and the following changes were made: addition of 1 mM KCN, having the gas-delivery tubing only contain N₂, and addition of sodium dithionite. [O₂] in the culture and OCR are shown. The shaded periods are where spectra are averaged over to give rise to the remission spectra. R is the reference period. **g** Difference attenuance spectrum showing the reduction of cytochromes that occurred as KCN was added (spectrum I *minus* spectrum R), as the cells go anoxic (II *minus* R), and full reduction with dithionite (III *minus* R). **h** Difference spectrum of the purified respiratory supercomplex (cytochrome *bcc:aa₃*).

The electron transport chain (ETC) builds the proton-motive force (PMF) in respiratory organisms, by translocating protons across an energy-transducing membrane. The PMF has two components, the seperation of charge ($\Delta\Psi$; 'membrane potential') and the imbalance of pH ($\Delta$pH). As ATP synthase consumes the PMF to phosphorylate ADP, it imposes respiratory control on the ETC through 'backpressure', the PMF-based 'force' that resists forward catalysis by the ETC, slowing the O₂-consumption rate (OCR). A schematic of the mycobacterial

bioenergetic system is shown in Fig. 1a. Remarkably, when bedaquiline is placed on cells an increase in OCR is observed, opposite to how an inhibitor of ATP synthase ought to act[14–16]. Two proposals were advanced to explain this effect. One is that bedaquiline is itself an ionophore, acting when 'tailgating' ATP synthase; this activity may be either bedaquiline acting as a protonophoric uncoupler[15], or an ionophoric uncoupler that interchanges $K^+$ and $H^+$ potentials[16]. The uncoupler proposals were supported by later cryo-EM studies reporting three kinds of binding sites in the $F_O$ domain and that bedaquiline disrupts the physical communication between the ATP synthase $F_O$ and $F_1$ domains as its concentration increases[9]. A second proposal is that mycobacteria either lack backpressure or have mechanisms to collapse it so that when ATP synthase is inhibited, the ETC operates at the same or increased rate[14]. A fundamental difficulty in differentiating between these activities is that an increase in OCR can be caused by a collapse in PMF, metabolic changes that increase electron delivery to the chain, or changes in the regulation of enzymes within the ETC. Therefore, studies on intact cells that measure OCR alone and come to contradictory conclusions are at a methodological impasse[14,15]. Moving to a simpler system to study bedaquiline action still requires an energy-transducing membrane, complicating experiments and providing more conflicting evidence. Inverted cytoplasmic membrane vesicles (IMVs) allow access to the active sites of respiratory enzymes whilst maintaining a proton-impermeable membrane, allowing respiratory substrate oxidation assays and qualitative PMF reporters to be used. However, IMVs provide inconsistent results with bedaquiline being found to uncouple succinate-energised IMVs[15], but not NADH-energised IMVs[14,17]. (Proteo)liposomes have been used to show that at μM concentrations bedaquiline dissipates the PMF and that this activity is enhanced by the presence of *E. coli* ATP synthase[16].

*M. tuberculosis* is a demanding organism to work with. It grows slowly, has a fastidious nature, and readily spreads through aerosols, making it a level 3 biosafety agent. For these reasons, *Mycobacterium smegmatis* has found wide use as a model mycobacterium for molecular physiology. A saprotrophic microbe that grows relatively quickly, for which a range of powerful genetic tools have been developed, has led to *M. smegmatis* being described as the 'vanguard of mycobacterial research'[18]. Supporting the utility of *M. smegmatis* for bioenergetic research, bedaquiline was discovered with this organism[6], and the respiratory complexes are highly conserved, as shown in high-resolution structures determined from both species, such as ATP synthase[9,10] and the *bd* oxidase (CydAB)[19,20]. *M. smegmatis* has a growing role as a bioenergetic model organism, for example, providing insights into aerobic $H_2$ and CO oxidation[21,22]. However, the physiology of an organism that divides every 4 h will be distinct from one doubling every ~24 h (or longer). For work closer to *M. tuberculosis* that is unsuitable for level 3 biosafety conditions, *M. tuberculosis* strains mutated to be less pathogenic, typically developed for vaccines, offer a suitable alternative. Bacillus Calmette–Guérin (BCG) strains are much less pathogenic but have accumulated extensive genomic changes over decades of culture[23]. To address this, a commonly used *M. tuberculosis* pathogenic cell line (H37Rv) has had the main inactivating mutation from BCG introduced (ΔRD1) alongside pantothenate autotrophy (ΔpanCD) to give rise to mc²6230 or mc²6030[24]. *M. tuberculosis* mc²6230 is therefore less infectious and carries a metabolic defect and is used here.

In this work, remission spectroscopy is used to measure cytochromes in living cells. Cytochromes (*cyto-*, cellular and *chrome*, colour) are haem-containing redox enzymes. Many of the ETC complexes are cytochromes as their haems act as integral mechanistic redox centres. Absorbing light strongly, and that absorbance depending on the oxidation state of the Fe atom, renders cytochromes excellent spectroscopic handles for electron transfer. Originally observed in intact organisms[25], the strong scattering of visible-wavelength light by cells means detailed spectroscopic measurement of cytochromes has

historically relied on fractionating cells to subcellular structures such as mitochondria[26,27] or isolated enzymes[28]. However, for intact cells, remission spectroscopy provides a method to measure spectra in highly-scattering cell suspensions[29]. Here, a stable light source illuminates the culture through a small aperture and the light that has been scattered 180° ('re-emitted') through another small aperture is detected. Because of the set-up, remission spectroscopy cannot provide absolute absorbance spectra relative to a spectroscopically clear reference but instead provides deflections from a baseline ('difference spectra'). Remission-spectroscopic measurements made on cell suspensions have a differential pathlength difference of < 3% from ~530 nm to 740 nm[30], which provides sufficiently accurate remission spectra that known spectra of isolated cytochromes can be additively combined to recreate the observed signal, attesting to the spectral fidelity of the approach[31]. A note on terminology: formally, all visible-wavelength spectroscopies measure attenuance, the depletion of light reaching the detector by both scattering or absorption of photons by an analyte, but over time absorbance has become the accepted, if incorrect, term. Here, we use attenuance and absorbance in their formal senses.

By considering bioenergetic principles and reviewing the literature, we formulate three hypotheses here for the mechanism of bedaquiline (also shown as a schematic in Fig. 1b–d). Activity 1- **uncoupler activity**: bedaquiline binding makes ATP synthase leaky to protons[9,15] or activates a proton carrier in the membrane[14] analogous to mammalian uncoupling proteins, in both cases having the functional effect of creating an uncoupler that collapses the PMF. Activity 2- $K^+/H^+$ **ionophore activity**: when bedaquiline binds to (tailgates) ATP synthase, a $K^+/H^+$ ionophore is created, which selectively interchanges ΔpH and transmembrane $K^+$ gradients[16]. Activity 3- **oligomycin-like activity**: bedaquiline binds to and jams ATP synthase like the canonical mammalian ATP synthase inhibitor oligomycin without additional effects; in this case how the cell responds may create additional activities[14,32]. We test these hypotheses by combining remission spectroscopy with OCR measurements to observe the mycobacterial ETC in vivo. By breaking the methodological impasse we find that bedaquiline resembles oligomycin at both nM and μM concentrations; the previously reported increase in OCR stems from activation of the *bd* oxidase (CydAB). We show this activation occurs on a time-scale of seconds, too fast for transcriptional or translational changes. A firm understanding of bedaquiline action in vivo provides a mechanistic basis for its effects on cellular metabolism and drug synergy, clarifies the importance of CydAB beyond $O_2$ scavenging in hypoxic conditions, and guides the development of future combination therapies against tuberculosis.

## Results

### A bioenergetic chamber allows simultaneous measurements of mycobacterial cytochrome redox state and oxygen consumption rate

To establish that our system can make measurements on mycobacteria, we placed *M. smegmatis* in the bioenergetic chamber, a schematic of which is shown in Fig. 1e. OCR increased gradually as the cells grew (Fig. 1f). The addition of KCN, a canonical inhibitor of $aa_3$-type oxidases, caused an increase in light attenuance and a drop in OCR, indicative of the oxidase being blocked and electrons accumulating in the ETC. The increase in attenuance comes from the haem groups bound in cytochromes. KCN predominantly caused an increase in the electron occupancy of the *a* and *c* cytochromes (600 and 555 nm respectively), which have the highest $E_m$ values (Fig. 1g), making them the easiest to reduce. Removing $O_2$ further increased the electron occupancy of the *c*-cytochromes and induced a large increase in the *b*-cytochromes (565 nm); adding the strong reductant dithionite causes a final increase in *b*-cytochrome electron occupancy. Comparing the remission spectra of cells and that of the isolated supercomplex (Fig. 1h) shows that the supercomplex is the dominant cytochrome.

## Bedaquiline causes a redistribution of electrons in the mycobacterial ETC

We examined the effects of bedaquiline on *M. smegmatis* by exposing a culture in the chamber sequentially to 30 nM, 300 nM, and 1.5 μM bedaquiline, causing a small increase in OCR (Fig. 2a), in line with previous observations[14,16]. A marked increase in the electron occupancy of the *b* and *c* cytochromes was observed in the remission spectrum (Fig. 2d and Supplementary Fig. 1), indicating a back-up of electrons in the ETC that saturates before 30 nM. We compared bedaquiline to the ethanol vehicle and found that while ethanol caused a small increase in electron occupancy, bedaquiline caused a far greater increase (Supplementary Fig. 2).

To determine whether these changes required the binding of bedaquiline to ATP synthase, the bedaquiline-resistant AtpE[D32V] point mutant was tested[6]. With the same concentrations of bedaquiline, the response was greatly diminished, with OCR only increasing in line with the growth of the cells in the chamber (Fig. 2b). The remission spectrum changes were decreased and no longer saturated (Fig. 2e and Supplementary Fig. 1), indicating that binding to ATP synthase is required for the effect of bedaquiline on the ETC. *M. tuberculosis* mc²6230 exposed to 120 nM bedaquiline behaved similarly (Fig. 2c, f; and Supplementary Fig. 1), demonstrating the effect is conserved in both mycobacterial species.

To unmix and quantify the individual cytochrome signals within the observed remission spectra, decomposition was applied (Fig. 2g)[33]. This approach separated the mixed spectra into individual cytochrome and background components (Fig. 2i, j). Interestingly, while bedaquiline caused a robust increase in electron occupancy of the $b_{557}$, $b_{564}$, and *c* cytochromes, *a* changed little upon the addition of bedaquiline, despite the signal appearing clearly when cells are taken anoxic (Fig. 1g). While there is insufficient mechanistic information about the mycobacterial oxidase to explain this observation, mammalian systems respond similarly to the ATP synthase inhibitor oligomycin[31].

## Resolving the mechanism of bedaquiline in living cells

Knowing that the bedaquiline-induced redistribution of electrons in the mycobacterial ETC is measurable and quantifiable in our system, we set out to systematically test the mechanistic hypotheses in Fig. 1 through comparison with known bioenergetic effectors.

We first tested for uncoupling (activity 1) by examining the action of the established uncoupler CCCP. CCCP passes through membranes in both its protonated and deprotonated forms, allowing it to carry protons through the membrane before returning, acting as a short-circuit to collapse both ΔΨ and ΔpH. In *M. smegmatis*, additions of 1 and 5 μM CCCP caused an increase in OCR (Fig. 3a), consistent with an uncoupler that decreases PMF, stimulating respiration. In contrast to bedaquiline (Fig. 2i), CCCP caused the electron occupancy of $b_{564}$ and *c* to decrease (Fig. 3e, i). This decrease occurred because the *b*-cytochromes are buried in the membrane and are sensitive to ΔΨ[34]. In *M. tuberculosis* mc²6230, the change in $b_{564}$ was not statistically significant but the electron occupancy of *c* significantly decreased; $b_{557}$ (which is not part of the respiratory supercomplex) increased (Fig. 3f, j). In both *M. smegmatis* and *M. tuberculosis* mc²6230, 25 μM CCCP caused a collapse in OCR, with the cytochromes having even lower electron-occupancy, presumably due to off-target effects, demonstrating the importance of titrating CCCP. These differences in response probably reflected differences in the pre-treatment electron-occupancies and turnover frequencies of the ETCs in *M. smegmatis* and *M. tuberculosis* mc²6230. In sum, CCCP and bedaquiline both stimulate an increase in OCR but have different effects on the electron-occupancy of the supercomplex-associated cytochromes $b_{564}$ and *c*, inconsistent with bedaquiline having an uncoupling effect.

We next tested the hypothesis that bedaquiline is a K⁺/H⁺ ionophore (activity 2 in the introduction) by examining nigericin to compare with the bedaquiline response. Nigericin is a canonical K⁺/H⁺ ionophore, possessing 24-fold selectivity for K⁺ over Na⁺[35]. K⁺ is abundant in the cytoplasm (~300 mM in *Bacillus subtilis*[36]; ~200 mM in *E. coli*[37]) compared to the low [K⁺] in the 7H9 growth medium (~10 mM). Therefore, nigericin will catalyse the efflux of K⁺, building ΔΨ, and the influx of H⁺, collapsing ΔpH. Interestingly, 1 μM nigericin has different effects on *M. smegmatis* and *M. tuberculosis* mc²6230. In *M. smegmatis*, there was a sudden drop in OCR and the electron occupancy of the *b* and *c* cytochromes decreased; after about 10 min, OCR recovered to near its starting point and the changes in the cytochromes began to revert (Fig. 3c, g, k). In *M. tuberculosis* mc²6230, OCR dropped and failed to recover, accompanied by a large but variable increase in *c*, and a decrease in $b_{564}$ electron occupancy (Fig. 3d, h, l). Currently, we cannot rationalise these changes as ion homoeostasis is little studied in mycobacteria. However, in both organisms, a drop in electron-occupancy of $b_{564}$, which is sensitive to ΔΨ, was observed. The opposite behaviour of *c* may reflect the more pronounced but transitory drop in OCR displayed by *M. smegmatis* compared to *M. tuberculosis* mc²6230. Whilst we lack a satisfactory explanation for these observations, they are completely distinct from the changes induced by bedaquiline, discounting the hypothesis that bedaquiline is a K⁺/H⁺ antiporter.

To corroborate these findings, we examined the effect of the covalent ATP synthase inhibitor DCCD on *M. smegmatis*. 10 μM DCCD behaved similarly to bedaquiline: it induced a small increase in OCR and an increase in the electron occupancy of the *b*- and *c*-cytochromes (Supplementary Fig. 3a–c). To measure the effects of DCCD and bedaquiline on ΔΨ and ΔpH we used an isotope accumulation assay on whole cells[15]. Treatment with bedaquiline or DCCD significantly increased ΔΨ relative to a DMSO control, while neither had a significant effect on ΔpH (Supplementary Fig. 3d, e), suggesting an overall increase in the PMF. This increase is consistent with bedaquiline inhibiting ATP synthase, blocking the major conduit of protons back into cells and increasing PMF.

## The *bd* oxidase (CydAB) acts as a 'relief valve' to protect mycobacteria from bedaquiline-induced backpressure

The lack of protonophoric or ionophoric activity displayed by bedaquiline left us with activity 3: oligomycin-like activity. To accept this activity, there needs to be a mechanism to explain the remarkable increase in OCR that bedaquiline induces, which defies the canonical understanding of how an ATP synthase inhibitor ought to act. One possibility is that in response to bedaquiline, flux is rerouted from the supercomplex to the *bd* oxidase, CydAB. Unlike the $aa_3$-type oxidase in the supercomplex (Cta), CydAB lacks vectorial proton pumping activity, it only builds the PMF as the 'chemical' protons for the reduction of $O_2$ to $H_2O$ most likely come from the cytoplasm, and electrons from the oxidation of $MQH_2$ move from the periplasm[19,20]. It is therefore much less sensitive to the PMF than the supercomplex, raising the possibility that if a high PMF slows the supercomplex electron flux can be redirected to Cyd to relieve reductive pressure. To test this hypothesis, we moved to strains with deletions of the *cydAB* operon in *M. tuberculosis*[38] and *M. smegmatis*[39]. We attempted to use the characteristic *d* cytochrome signal to detect changes to CydAB at 630 and 650 nm[19,20,40] using a wide spectrum but the changes were too small compared to noise to be reliably interpreted (Supplementary Fig. 4a, b). However, spectra of isolated cytoplasmic membranes showed clear levels of CydAB in *M. smegmatis* grown aerobically at pH 6.6 (Supplementary Fig. 4e, g), which was decreased in Δ*cydAB* (Supplementary Fig. 4f, h). There is a second *bd* oxidase (AppCB) present in *M. smegmatis* but not *M. tuberculosis*[41], which may explain the small *d* cytochrome present in *M. smegmatis* Δ*cydAB*.

When *M. smegmatis* Δ*cydAB* was exposed to 30 nM bedaquiline there was an immediate decrease in OCR, whilst the electron occupancy of *b* and *c* cytochromes increased as before (Fig. 4a, e). This behaviour is similar with the behaviour of oligomycin acting on

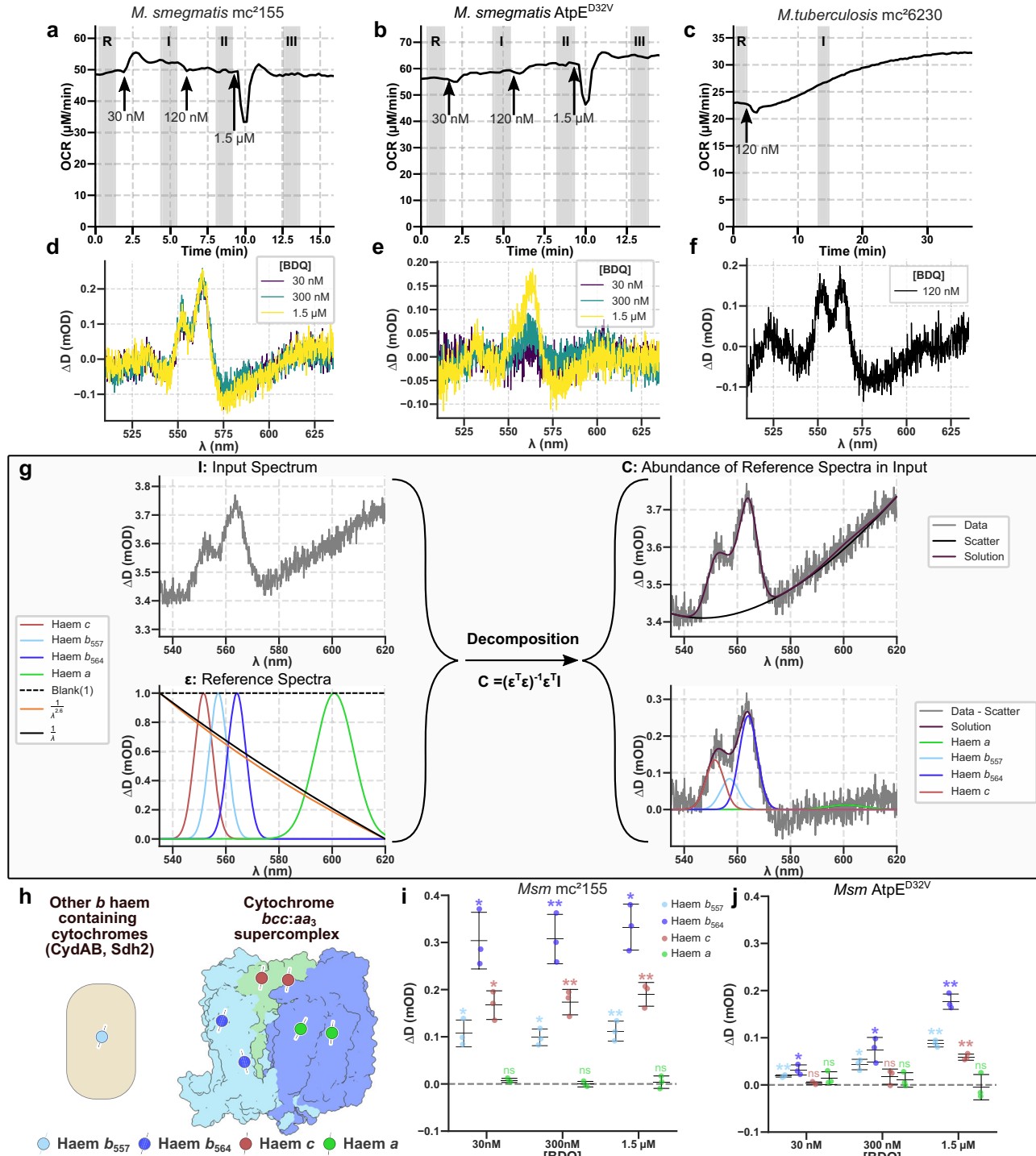

**Fig. 2 | The effects of bedaquiline on *M. smegmatis* and *M. tuberculosis* mc²6230 measured in the bioenergetic chamber. a–c** OCR traces for *M. smegmatis* mc²155, *M. smegmatis* AtpE[D32V], and *M. tuberculosis* mc²6230 responding to the addition of bedaquiline. **d–f** Representative visible-wavelength difference remission spectra from (**a–c**); spectra were averaged over the shaded periods and subtracted to give the presented spectra following a further subtraction of a straight line. **g** Example of the spectral decomposition model and approach used here, showing the input spectrum and the reference spectra. On the right, the background components are shown on the input spectrum, demonstrating a reasonable fit. Below are input

spectra with the background components subtracted, to clearly show the fit between cytochrome signals and the data. **h** Cartoon representation of cytochrome complexes corresponding to fitted signals. **i, j** Results of signal fitting on the above runs providing individual cytochrome changes. Cells were suspended in 7H9 + 50 mM glycerol; 0.2 mM pantothenate was added in experiments with *M. tuberculosis* mc²6230 and 5 mM D-(-)-arabinose for experiments with *M. smegmatis*. **a–f** Are representative replicates from at least three independent experiments. Mean is shown ± SD ($n = 3$). Comparisons were made using a one sample Student's $t$-test. *$p \leq 0.038$, **$p \leq 0.01$, ns $p \geq 0.05$.

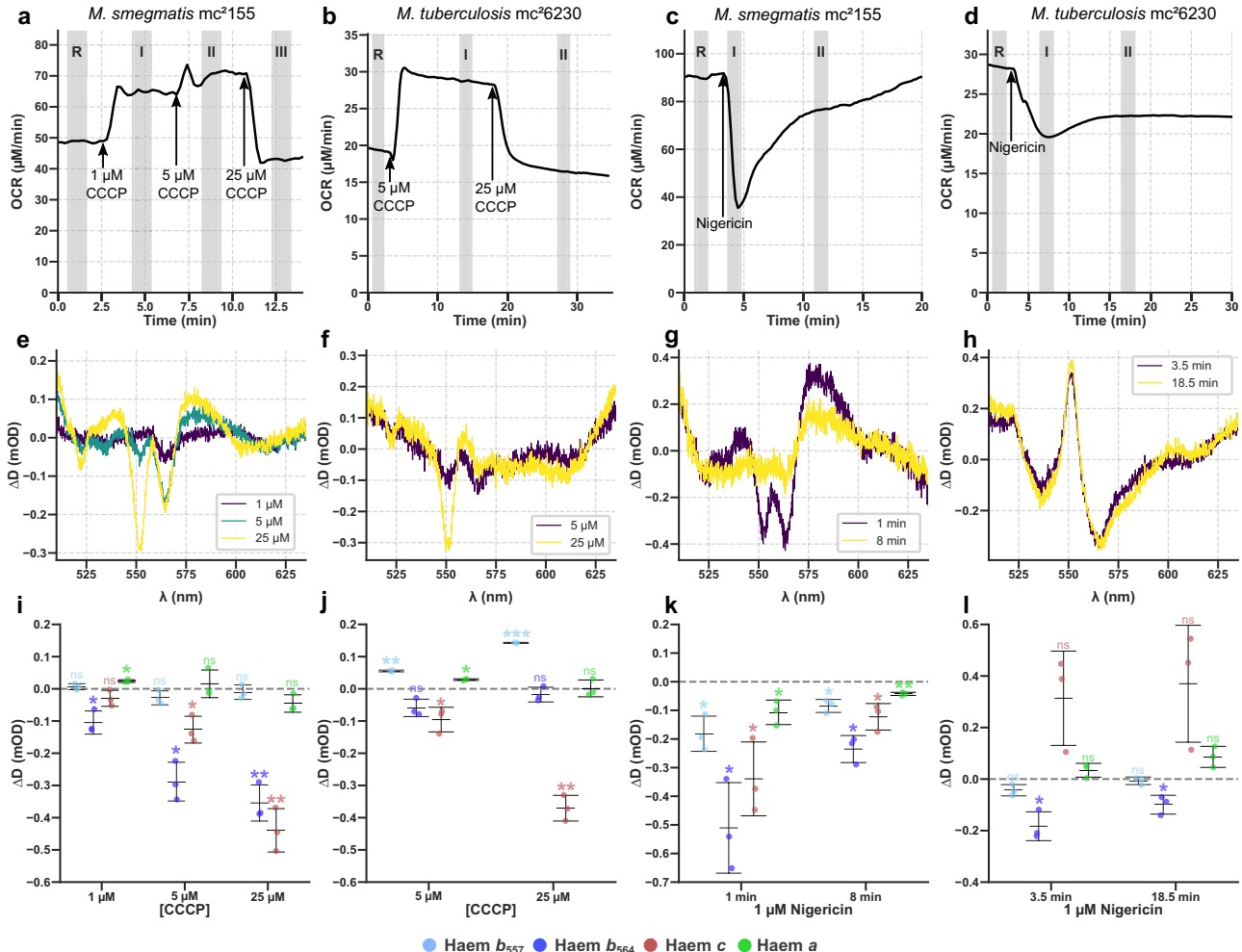

**Fig. 3 | Effects of the canonical uncouplers CCCP and nigericin on myco-bacterial OCR and remission spectra.** The effects of CCCP and nigericin on mycobacterial OCR (**a**–**d**) and remission spectra (**e**–**h**). The effects of CCCP and nigericin on mycobacterial OCR (**a**–**d**) and remission spectra (**e**–**b**). Data is presented for how CCCP affected *M. smegmatis* (**a**, **e**) and *M. tuberculosis* mc²6230 (**b**, **f**) and how nigericin affected *M. smegmatis* (**c**, **g**) and *M. tuberculosis* mc²6230 (**d**, **h**). **i**–**l** Individual cytochrome changes resulting from quantification of the spectral

changes for the panels above using the Gaussian unmixing model for three independent experiments. Cultures were concentrated to an $OD_{600}$ of -1.8 (*M. smegmatis*) and -5 (*M. tuberculosis* mc²6230) for analyses. The oxygen concentration in the chamber was held at 100 μM. Cells were suspended in 7H9 + 50 mM glycerol; 0.2 mM pantothenate was added in experiments with *M. tuberculosis* mc²6230. Mean is shown ± SD (*n* = 3). Comparisons were made using a one sample Student's *t*-test. *$p \leq 0.049$, **$p \leq 0.01$, ns $p \geq 0.05$.

mitochondria, demonstrating that CydAB mediates the increase in OCR in response to bedaquiline. However, in *M. tuberculosis* Δ*cydAB* OCR still increased, but much less than in *M. tuberculosis* mc²6230 (Fig. 4b). We speculated this was due to one of two reasons: (1) bedaquiline is less capable of reaching its target site in *M. tuberculosis* mc²6230 through differences in bedaquiline influx or efflux or (2) the bioenergetic system of *M. tuberculosis* has additional mechanisms to compensate for the backpressure induced by bedaquiline compared to *M. smegmatis*. In support of (1), we noticed that *M. tuberculosis* responded much more slowly to bedaquiline than to other small molecule effectors like nigericin and CCCP (Fig. 2c and Fig. 3b, d). If the differences were due to the ability of bedaquiline to reach its target site, increasing the concentration of bedaquiline should overcome it. When the concentration of bedaquiline was increased to 10 μM, there was a robust decrease in OCR in both *M. smegmatis* and *M. tuberculosis* Δ*cydAB* (Fig. 4c, d). The differences observed may have been due to the differences in expression of the MmpS5-MmpL5 transporter, which is implicated in bedaquiline efflux[42].

Taking bedaquiline from nM to μM concentrations also tests the hypothesis that only at μM concentrations does bedaquiline act as an uncoupler, either as a small molecule proton transporter or at ATP

synthase[9,16]. The remission spectra of an uncoupler would have been expected to resemble that of CCCP as bedaquiline concentration was increased to 10 μM, indicating a decrease in electron occupancy. Instead, no significant change in *M. smegmatis* or *M. tuberculosis* mc²6230 (Fig. 4 and Supplementary Fig. 5) was observed, demonstrating that bedaquiline lacks uncoupling activity on intact cells, even at high concentrations.

### Slowing the supercomplex directly induces electron flux redirection to CydAB

If slowing the activity of the supercomplex causes an increase in OCR through redirection of electron flux, CN⁻ should also induce the same effect but independently of ATP synthase. We presumed that in Fig. 1, so much KCN was added that CydAB was also inhibited as it is resistant but not immune to CN⁻[43], so the increase in OCR was greatly dampened. Therefore, we explored exposure of *M. tuberculosis* mc²6230 and Δ*cydAB* to a larger range of KCN concentrations: 100 μM, 1 mM, and 10 mM (Fig. 5). At 100 μM, OCR in *M. tuberculosis* mc²6230 increased while Δ*cydAB* displayed a decrease; in both cases the electron occupancy of the cytochromes increased, supporting our model for the redirection of electron flux. Increasing KCN concentrations to 1

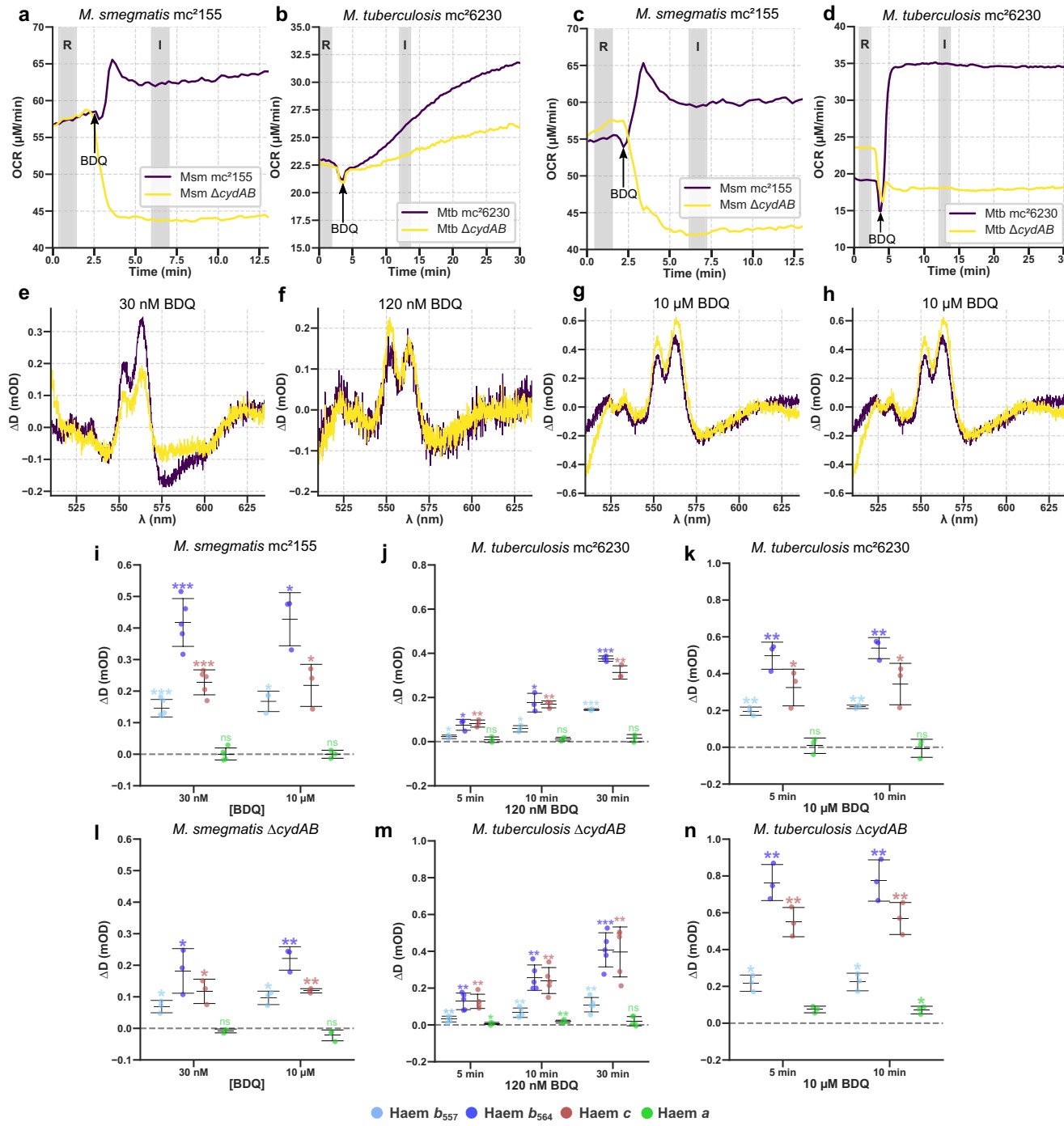

**Fig. 4 | The effects of nanomolar and micromolar concentrations of bedaquiline on *M. smegmatis* and *M. tuberculosis* mc²6230. a–d** OCR of *M. smegmatis* mc²155 and Δ*cydAB* and *M. tuberculosis* mc²6230 and Δ*cydAB* responding to nM and μM concentrations of bedaquiline. **e–h** Remission difference spectra for cultures in (**a–d**). **i–n** Bedaquiline induced cytochrome changes for *M. smegmatis* mc²155 and Δ*cydAB* and *M. tuberculosis* mc²6230 and Δ*cydAB*. b and f include data from Fig. 2c, f for comparison. Values for changes in attenuance and OCR were recorded 3 min post-addition for *M. smegmatis*, 10 for *M. tuberculosis*. Mean is shown ± SD ($n = 3$ except *M. smegmatis* mc²155 responding to 30 nM bedaquiline (**i**) and *M. tuberculosis* Δ*cydAB* responding to 120 nM bedaquiline (**m**), where $n = 5$). Comparisons were made using a one sample Student's *t*-test, where *$p \le 0.047$, **$p \le 0.01$, ***$p \le 0.001$, and ns $p \ge 0.05$.

and 10 mM caused OCR to drop, as CydAB begun to be inhibited[43]. Interestingly, at 10 mM, the supercomplex-associated $b_{564}$ cytochrome decreased in electron occupancy in *M. tuberculosis* mc²6230 but not in Δ*cydAB*. The supercomplex *b*-haems are hard to fully reduce[44] and the presence of ΔΨ will help. At 10 mM KCN, the supercomplex is fully inhibited so the PMF (and ΔΨ) collapses but CydAB is still sufficiently active to transfer electrons; in Δ*cydAB* this cannot happen so $b_{564}$ stays reduced.

## Human mitochondria are unaffected by bedaquiline

Our results have shown that the canonical relationship between ATP synthase activity and OCR is not as reliable as previously thought. While prior studies found that the addition of bedaquiline leaves OCR in mammalian cells unchanged[14,45], recent work found that bedaquiline inhibits the in vitro activity of human mitoplasts and purified ATP synthase with an IC₅₀ of ~500 nM[10,11], below concentrations found in patient sera (~1–10 μM)[46]. To determine whether bedaquiline was

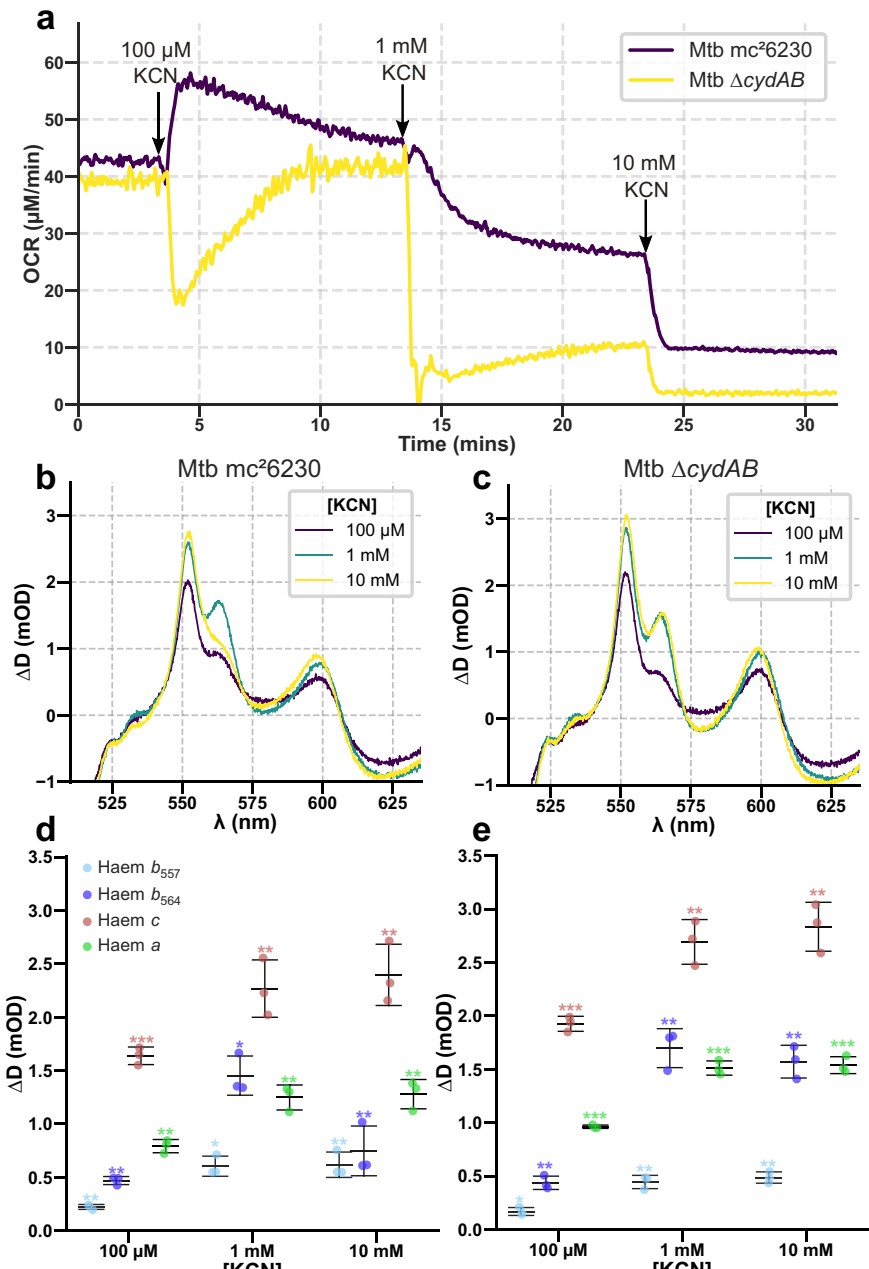

**Fig. 5 | KCN inhibition of the supercomplex redirects electron flux to CydAB in** ***M. tuberculosis*** **mc²6230. a** OCR of *M. tuberculosis* mc²6230 (purple) and *M. tuberculosis* mc²6230 Δ*cydAB* (yellow) treated with μM and mM doses of KCN. **b, c** Difference spectra of *M. tuberculosis* mc²6230 and Δ*cydAB*, using pretreatment as a baseline. Traces are representative replicates from at least three independent experiments. **d, e** Changes in individual cytochrome signals following treatment with KCN. Mean is shown ± SD ($n = 3$). Comparisons were made using a one sample Student's $t$-test, where $*p \leq 0.031$, $**p \leq 0.01$, and $***p \leq 0.001$.

capable of inhibiting human mitochondrial function, the effects of bedaquiline on the HEK293T cell line were tested. Addition of bedaquiline at 10 μM yielded no differences in remission spectra or OCR compared to no addition or ethanol (the vehicle) only; in contrast, the canonical inhibitor oligomycin A caused an increase in both *b* and *c*-cytochrome electron occupancy (Fig. 6; Supplementary Fig. 6). Therefore, we observed no evidence for bedaquiline inhibiting mammalian cells whilst inducing adaptive mechanisms analogous to those found in mycobacteria. Combined with observations that bedaquiline is found in mitochondria at low concentrations compared to other parts of the cell[47] we conclude that bedaquiline either does not enter mitochondria, or enters and is rapidly pumped out, and therefore leaves mammalian ATP synthase unaffected.

## Discussion

Of the three possible hypotheses set out in Fig. 1, our results are inconsistent with bedaquiline acting as an uncoupler or ionophore (activities 1 and 2). Our model for action is as follows: bedaquiline binds to the ATP synthase c-ring, stopping catalysis. As catalysis is stopped, the major conduit of H⁺ from the periplasm to the cytoplasm is blocked, increasing PMF (backpressure) that resists forward-catalysis by the ETC. As the respiratory supercomplex is strongly PMF-coupled, its catalysis is slowed, decreasing OCR. However, CydAB is rapidly activated, allowing OCR to increase beyond pre-bedaquiline levels due to electron flux redirection.

 Previously, the nature of backpressure within mycobacterial bioenergetics had been questioned, with proposals that it is either not

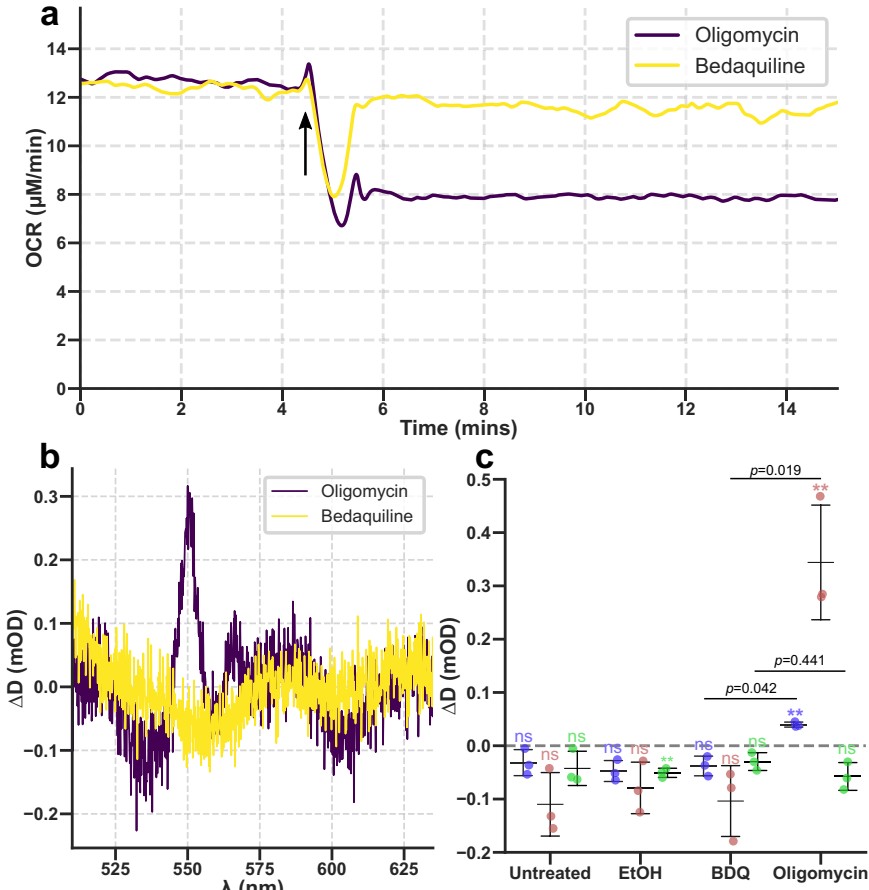

**Fig. 6 | Human mitochondria within living suspension cells are unaffected by bedaquiline but are affected by oligomycin.** Changes in OCR (**a**), remission spectra (**b**), and individual cytochromes (**c**) for human HEK293T cells exposed to 10 μM bedaquiline or 1.5 μM oligomycin. Arrow indicates time of drug addition. Spectra were recorded 5 min after the addition of bedaquiline or oligomycin. $n = 3$.

Error bars are ± standard deviation. Comparisons were made using a one sample Student's $t$-test. **$p \leq 0.01$, ns $p \geq 0.05$. For comparisons between conditions, a Welch's $t$-test was used with a Bonferroni correction for multiple comparisons, with adjusted $p$ values labelled.

sensed or collapsed by energy spilling mechanisms[14]. By directly measuring the status of the ETC in vivo, we find that bedaquiline increases backpressure and that the cytochromes of the ETC directly sense it. Our $\Delta cydAB$ experiments in both *M. tuberculosis* mc²6230 and *M. smegmatis* show that any induction of an energy spilling mechanism, beyond redirection of electron flux, must either be absent or have a minimal effect on PMF. This backpressure explains why bedaquiline induces increased succinate accumulation and elevated NADH/NAD⁺ levels[12–14], signatures of high PMF and backpressure in mammalian systems[47,48].

The *bd* oxidases such as CydAB are traditionally seen as $O_2$ scavenging enzymes[40]. However, their affinity for $O_2$ does not explain their role in pathogenesis: the apparent $K_M$ of the mycobacterial respiratory supercomplex for $O_2$ is ~0.9 μM[49], more than sufficient to mediate growth at $O_2$ concentrations below the ~2 μM $O_2$ found within granulomas[50]. CydAB compensates for loss of the supercomplex during infection[51], but its specific function within the branched mycobacterial respiratory chain is unclear. Here, we find a role for CydAB as a relief valve for reductive pressure at plentiful $O_2$. This function also explains the role of CydAB in acid tolerance, where *M. tuberculosis* experiences reductive pressure[52,53].

The menaquinone-pool is a central metabolic hub but is invisible to our spectroscopic approach. The bedaquiline-induced slowing of the supercomplex, which caused the supercomplex to fill with electrons, occurred simultaneously with the activation of CydAB that emptied the menaquinol pool of electrons. Therefore, the redox-state of the menaquinone pool was unclear. However, the supercomplex

associated $b_{564}$ signal acts as an indicator. The $E_m$ values of the haems that make up $b_{564}$, $b_L$ and $b_H$, in the closely related supercomplex from *Corynebacterium glutamicum* are more negative than those from ubiquinol-reactive $bc_1$ enzymes to enable efficient electron transfer[44]. Therefore, the assumption commonly made for mammalian enzymes, that the $b$-haems of $bc_1$-family enzymes is close to or at equilibrium with the Q-pool[29,54], is reasonable here and we tentatively conclude that the increase in electron occupancy we saw in $b_{564}$ in *M. tuberculosis* mc²6230 and *M. smegmatis* means the menaquinone pool is probably filling with electrons overall, which will affect the many enzymes that react with it[55].

We lack a verified mechanism for how CydAB is activated but there are hints in the literature. As CydAB does not pump protons, the overall $\Delta G$ over the enzyme will be strong (-900 mV, -86 kJ mol⁻¹) and it must be under kinetic control. Any mechanism needs to occur on a second time scale. Possible options include a regulatory switch and allostery; the two are not mutually exclusive. A disulphide bond next to the periplasmic quinol binding-site was observed in structures of mycobacterial CydAB[19,20,56]. Reducing agents that break the disulphide decrease activity in vitro, possibly creating a link to ROS regulation[56]. CydAB may also be tuned to react with menaquinol so that when the Q-pool fills with electrons, menaquinol binds to CydAB at an allosteric site and activates it. Recombinant CydAB can now be produced in sufficient quantities for detailed kinetic analysis[56,57], raising the possibility of further detailed investigations on the isolated enzyme.

Our results provide insights into bedaquiline tolerance and antimicrobial synergy. First, the capacity of CydAB to relieve bedaquiline-induced backpressure provides a mechanism for its role in bedaquiline tolerance[15,32]. Second, the shift in flux from the supercomplex to CydAB explains the lack of synergy between supercomplex inhibitors and bedaquiline[58] or its analogues[59,60]. A CydAB inhibitor could mitigate these challenges as demonstrated by their enhancement of the bactericidal activity of bedaquiline and supercomplex inhibitors[38,61]. Finally, we can explain the synergy between clofazimine and bedaquiline[14,62]. Clofazimine is reduced by NDH-2, whereupon it redox cycles, creating ROS[63]. As measured in our remission spectra, bedaquiline-induced backpressure causes the electron occupancy of the *b* and *c*-haems to increase and most likely also the menaquione-pool. This backpressure-induced reduction will in turn reduce NDH-2, increasing its side-reaction with clofazimine and generation of ROS.

Our observations around $O_2$ consumption and bedaquiline are consistent with Lamprecht et al[14], with a single exception: that deletion of *cydAB* stops the bedaquiline-induced increase in OCR. That study used a strain of *M. tuberculosis* nominally deleted in *cyd*[64], but the strain was later revealed to only harbour a small deletion of *cydB*, all of *cydD*, and much of *cydC*, leaving *cydAB* mostly intact[65]. CydDC shares its operon with CydAB but does not constitute the oxidase; in *E. coli* CydDC is a haem transporter and it is unclear if deleting CydDC affects CydAB function[66]. In this work, we used an independently generated deletion of the full CydAB open reading frames[38,39], avoiding this caveat. As an aside, we note that this observation is only possible is due to commendable decision of Arora et al to publish a clarification of their prior work[65].

Measurements of the bedaquiline-induced increase in OCR by other workers have found its magnitude to be larger than ours but this may be due to the inclusion of a carbon-source-free 'starvation' step before bedaquiline addition. For example, a suspension culture measurement of *M. smegmatis* respiring within a Clark electrode, similar in approach to our measurements here, displayed a ~100% increase in OCR but was made on cells that had been resuspended in carbon-free phosphate-buffered saline before glycerol was re-introduced and bedaquiline introduced[15]. Measurements using a XF96 Extracellular Flux Analyser on a monolayer of *M. tuberculosis* H37Rv cells found an even greater increase: ~600%[14]. These were made on cells that had been starved in carbon-free medium for 24 h. Our approach relied on taking mid-exponential phase bacteria, washing them in detergent-free growth medium so they remain energised, and quickly placing them in the chamber. Therefore, they should have remained in the mid-exponential growth phase when we conducted our experiments. We conclude the magnitude of the bedaquiline-induced increase in OCR is dependent on exact growth and measurement conditions.

Previous work relied on combining OCR measurements on intact bacilli with measurements on inverted membrane vesicles[14–16]. The advantage of our approach is that a single bacterial culture provides sufficient information to discern the mechanism of the added bioenergetic-effectors, avoiding effects arising from differences between intact cells, IMVs, and (proteo)liposomes. IMVs have long been a key bioenergetic experimental system but they only partially resemble cells. Whilst they sustain a PMF across their membrane, the polarity of the inner phase is reversed (i.e., the negative and alkali cytoplasm/matrix becomes the positive and acidic IMV/SMP lumen) and the membrane area-to-volume ratio differs by orders of magnitude (~100 $\mu m^2$ $fL^{-1}$ in IMVs vs ~0.2 $\mu m^2$ $fL^{-1}$ in bacteria) (Supplementary Fig. 7). Protonophoric agents require both the charge-driven movement of the ionic form and the neutral diffusion of the uncharged form across the membrane to occur at a sufficient rate but the balance of which is limiting is highly dependent on the system and state[67]. As such, the chemical differences between bedaquiline and FCCP/CCCP may explain why bedaquiline uncouples IMVs and not intact cells: bedaquiline has a pKa of 8.9, with neutral and positively charged forms while FCCP and CCCP have pKa values of 6.2 and 5.6 respectively with neutral and negative forms. FCCP/CCCP effectively uncouple intact bacteria but are relatively poor uncouplers of IMVs; e.g., fully uncoupling mammalian IMVs requires the use of gramicidin[68]. As the charge states of bedaquiline are reversed, most likely it uncouples IMVs reasonably but not intact cells.

Finally, we note that the study of bioenergetics has historically relied on combining quantitative measurements made on purified or minimal systems and cruder measurements on intact cells. Our approach allows quantitative measurements to be made in living mycobacteria and offers a clear mechanism for bedaquiline and insights into mycobacterial physiology. Cytochromes are universal in nonparasitic $O_2$-consuming organisms, allowing this approach to be applied broadly in the future.

## Materials and Methods

### Materials and strains

Bedaquiline, CCCP, nigericin, DCCD, and oligomycin A were purchased from AdooQ Bioscience, Alfa Aesar, Cayman Chemical Company, Sigma-Aldrich, and MedChemExpress, respectively. For chamber experiments these compounds were dissolved in EtOH and for isotope accumulation assays were dissolved in DMSO.

*M. smegmatis* strains used are mc²155 WT, AtpE$^{D32V}$ and Δ*cydAB*. AtpE$^{D32V}$ is mc²155 with a point mutation in the *atpE* gene conferring bedaquiline resistance[6] and Δ*cydAB* is mc²155 with deletion of the *cydAB* gene[39]. *M. tuberculosis* strains used were H37Rv ΔRD1 Δ*panCD* (mc²6230)[24] the Δ*cydAB* strain used in this study is *M. tuberculosis* mc²6230 with a deletion of the Δ*cydAB* locus[38].

### Growth conditions

*M. smegmatis* strains (mc²155, Δ*cydAB* and AtpE$^{D32V}$) were grown in Middlebrook 7H9 media (Sigma-Aldrich) supplemented with Tween 80 (0.05 % v/v) and glycerol (50 mM). For the AtpE$^{D32V}$ strain, 5 mM D-(-)-arabinose was added to reduce clumping. Media and supplements, save for D-(-)-arabinose, were combined prior to autoclaving. 7H10 agar plates supplemented with glycerol (68 mM) were inoculated with a −70 °C glycerol stock and incubated at 37 °C for ca. 72 h. Once single colonies were visible, plates were stored at 4 °C and single colonies were used to inoculate 50 mL of media in 250 mL conical flasks to prepare starter cultures. These cultures were incubated (37 °C, 200 rpm) until an $OD_{600}$ of 2.5–5 was reached (ca. 72 h).

*M. tuberculosis* strains (mc²6230 and Δ*cydAB*) were grown in Middlebrook 7H9 media (Sigma-Aldrich) supplemented with Tween 80 (0.05 % v/v), glycerol (55 mM), Middlebrook OADC growth supplement (bovine albumin fraction V (50 g/L), D-glucose (20 g/L), catalase (0.04 g/L), oleic acid (0.5 g/L), sodium chloride (8.5 g/L)) (10 % v/v) and pantothenate (48 mg/L). Tween 80 and glycerol were added prior to autoclaving with other supplements being added after. 5 mL of media in 25 mL vented flasks were inoculated from a −70 °C glycerol stock and incubated at 37 °C for 3-4 weeks.

HEK293T cells (ATCC CRL-3216) were cultured at 37°C in phenol-red free DMEM (Gibco) supplemented with 10% v/v foetal bovine serum (Gibco) in a 5% CO2 incubator.

### Preparing cells for measurements in the bioenergetic chamber

Initial experiments indicated that washing cells without detergent lowered the concentration at which the effect of metabolic effectors could be observed in the system, presumably due to partitioning of lipophilic compounds into the detergent. 50 mL *M. smegmatis* cultures were grown to mid-exponential phase in an orbital shaking incubator in 250 mL conical flasks. 3–5 mL of culture was pelleted in a swinging-bucket centrifuge at 3894 × *g* for 5 mins at 37 °C. Cells were then washed and resuspended in 5 mL growth medium to an $OD_{600}$ ~1.5. *M. tuberculosis* mc²6230 was grown in 40 mL cultures in vented T175 flasks to an $OD_{600}$ between 0.7–1.0. 35 mL of culture was pelleted in a

swinging-bucket centrifuge at 3894 × *g* for 5 mins at 37 °C. Cells were then washed and resuspended in 5 mL of growth medium without detergent or OADC to an OD$_{600}$ of ~6. Given the approximate OD values above, the normalised OCR was ~40 µM/min/OD for *M. smegmatis* and ~3 µM/min/OD for *M. tuberculosis* mc²6230.

HEK293T cells were grown in T175 flasks to ~70–80% confluency. Cells were harvested by treatment with 0.05% Trypsin-EDTA (Gibco). Harvested cells were pelleted for 3 min at 500 × *g* and resuspended to a final concentration of 1.0 ×107 cells/mL in FluoroBrite DMEM (Gibco), supplemented with 4 mM L-Glutamine and 25 mM HEPES pH 7.4. 5 mL of this suspension was used for chamber measurements.

## Bioenergetic chamber measurements

Measurements were made using a prototype of the Iberius Cell Spectroscopy System (CellSpex Ltd); many components were transferred from a system previously used to make measurements on mammalian cells[29,48]. The bioenergetic chamber is composed of three systems integrated together: a culture vessel, a remission spectroscopy system, and an oxygenation system. These are described in turn but work together simultaneously.

The culture vessel is composed of a 5 mL quartz crucible set into an aluminium block. Optodes for [O$_2$] measurement and remission spectroscopy are inserted through the crucible to access the bacterial culture. A magnetically-driven glass-covered stir bar run at 700 rpm is used to mix the culture and the system is maintained at 37 °C thermoelectrically.

For remission spectroscopy, a warm white light emitting diode (Luxeon-CZ 4000K-90 run at 300 mA) was used to illuminate the culture through a 2.0 mm NA0.5 POF optical fibre. Reemitted light back-scattered from the culture was collected with a 1.0 mm NA0.37 silica fibre located 10 mm below the illumination fibre. Previously the pathlength of such a setup has been measured as ~39 mm, using the 2nd deferential techniques of the 740 nm absorption band of water[31]. The collected light was fed into a Triax 320 spectrographs (Horiba) equipped with a DV401BV back-thinned charged coupled device (CCD; Andor Technology). Each spectrum was collected for 6 ms and 50 were averaged together every 0.5 s to give each working spectrum. For narrow spectra, the spectrograph was set to detect over the wavelength ranges 509- 640 nm, with a 600 groove / mm grating blazed at 500 nm and slit widths set to 200 microns. For the wide spectra, the spectrograph was set to detect over the wavelength ranges 481 - 751 nm, with a 300 grooves / mm at blazed 500 nm and slit widths set to 100 microns. For wide spectra, it was necessary to subtract the contribution from the optode phosphorescence at 650 nm, so the CCD collected 2 phases, phase one where the LED was on, and phase 2 where the LED was off. The output spectrum was then phase 1 minus phase 2. The spectrograph was calibrated daily against mercury emission lines (546.074 nm, 576.960 nm, and 579.070 nm). The spectra resolution in both cases was ≈1 nm.

The oxygenation system relies on measuring dissolved O$_2$ and delivering oxygen, via gas-permeable silicone tubing (Renasil©, Braintree Scientific) immersed in the suspension, at a rate equal to the rate consumed by the cells such that the oxygen concentration is constant. The oxygen consumption was then calculated from the rate of oxygen delivery corrected for the rate of change of oxygen concentration in the chamber. The rate of oxygen delivery was controlled by changing the N$_2$/O$_2$ gas mixture flowing through the tubing using a gas blender, and a feedback circuit varied the gas mixture to maintain the dissolved oxygen concentration at a set value. Oxygen tension was measured from the phase shift from a phosphorescent membrane (oxygen optode) using a phase fluorimeter (CellSpex Ltd). When compounds were added from cold ethanol stocks, there was typically a spike in OCR, occurring because of the O$_2$ dissolved in the stock solution, before a new steady state is set. When OCR changes rapidly, often the spike is obscured. In either case, rates were measured only

after the spike had settled. The optode was calibrated every few weeks to ensure accuracy and the permeability of the tubing was calibrated before every experiment.

## Spectral and statistical analysis

In a remission geometry, attenuance (D) is defined as:

$$D = -log_{10}\left(\frac{I}{I_0}\right)$$

Where I is the detected light intensity as a function of wavelength and I$_0$ is the illumination light intensity as a function of wavelength.

The changes to spectral attenuance induced by inhibitors or other changes were measured as deflections (difference spectra) from a reference period. The average attenuance (D) over the reference period was subtracted from the period of interest to give ΔD spectra. Typically, the periods were 0.5–2 min long to give sufficient signal for interpretation. For the display of ΔD spectra, a straight line is subtracted to account for changes to the baseline.

For unmixing, cytochrome and background signals from the observed mixture in the ΔD remission spectrum we used decomposition with model cytochrome signals composed of Gaussian functions. The observed *b*, and *c* peaks are each composed of a number of individual cytochromes, changing independently of each other ($c = c_1$ and $c_2$ in the respiratory supercomplex, *b* = two *b*-cytochromes in the Qcr component of the respiratory supercomplex, two in CydAB, two in Sdh2, and one in EtfD). Signals for cytochrome *c*, cytochrome $b_{564}$ and cytochrome *a* were included based on the spectrum of the purified supercomplex. To account for the other *b*-cytochromes, a signal centred at 557 nm was included and gave a solution that accounted for nearly all of the observed signal, judged by the flat residual spectrum.

To determine the magnitude of spectral signals, the difference spectra were fitted using a linear combination of Gaussian peaks and background functions (1, $1/\lambda^{2.6}$, $1/\lambda$). This was performed with decomposition using Eq. 1

$$C = \left(\varepsilon^T \varepsilon\right)^{-1} \varepsilon^T \Delta D \tag{1}$$

where **C** is a column matrix containing the contribution of each of the components. **ΔD** is a column matrix containing the attenuance difference at each wavelength and ε is a matrix containing each of the gaussian or background components.

Each of the cytochrome functions is described by Eq. 2:

$$y = e^{-Ln(2)\left(\frac{\lambda-\lambda_0}{w/2}\right)^2} \tag{2}$$

Where *c* is the centre of the peak and *w* is the full width at half maximum. For *M. smegmatis* and *M. tuberculosis*, the values for *c* and *w* for each peak is as follows, cytochrome *c* (551.5, 8), cytochrome $b_{557}$ (557.0, 8), cytochrome $b_{564}$ (564.0, 8) and cytochrome *a* (600.7, 17). For *H. sapiens*, the values were cytochrome *c* (550.5, 8.5) cytochrome *b* (564.0, 9) and cytochrome *a* (600.7, 17). All the components, peaks and background functions, were normalised across the fitted wavelength range such that the maximum was 1 and the minimum was 0.

To estimate when the system is detecting and fitting an individual cytochrome signal above noise we use a one-sample T-test[69]. The ΔD spectrum of the reference period is 0 so the null hypothesis (H$_0$) is that there is no difference in the data from 0 and the alternative hypothesis (H$_A$) is that there is.

$$t = \frac{\bar{Y} - \mu_0}{SE_{\bar{Y}}} \tag{3}$$

Where t is the t-statistic, Ȳ is the mean, μ$_0$ is the mean proposed by H$_0$, in this case 0, and SE$_{\bar{Y}}$ is the standard error.

For comparisons between conditions a 2-tailed T-test was used, assuming unequal variance (Welch's T-test). When needed, Bonferroni correction for multiple comparisons was applied to the *p* values needed to ascribe significance to results.

### Isolation of the 3xFLAG QcrB respiratory supercomplex (cytochrome *bcc:aa₃*)

As in Yanofsky et al, a *M. smegmatis* strain with a 3xFLAG tag at the C-terminus of the *qcrB* subunit of the supercomplex was generated with the ORBIT method[70,71]. The *M. smegmatis* strain was cultures as before except the starter culture was used to inoculate 1 L cultures in 2.5 L baffled flask (200 rpm 37 °C). These were grown until an $OD_{600}$ of 0.8 was reached at which point they were harvested via centrifugation (5000 $g$, 20 mins, 4 °C, SLC-6000 rotor). The cells were then homogenised in ~5 mL of chilled lysis buffer (50 mM Tris-SO4 pH 7.4 (RT), 150 mM NaCl, 5 mM EDTA, 1 mM MgCl₂) per g of cell weight and frozen until required.

After thawing and the addition of protease inhibitor tablets (Roche) (1 per 50 mL) and DNase (1 mg/mL), the cells were broken with 3 passages at 40 kpsi through a cell disruptor (Constant Systems). Unbroken cells and debris were pelleted by centrifugation (48k × $g$, 12 mins, spin, 4 °C, SS-34 rotor), followed by a longer ultracentrifugation spin to pellet the membranes (150 k$g$, 2 hrs, 4 °C, Type 45Ti rotor). These membranes were washed by resuspending and pelleted via an identical spin. The washed membranes were then resuspended at ~10 mg/mL to then be stored at −70 °C.

To isolate the 3xFLAG QcrB *bcc:aa₃*, membranes were thawed and diluted to 5 mg/mL, as determined by a BCA assay, using the purification buffer (50 mM Tris-SO4 pH 7.4 (RT), 150 mM NaCl). They were then incubated with 1 % DDM for 40 mins at 4 °C with agitation. Unsolublised material was then removed via ultracentrifugation (150k × $g$, 30 mins, Type 70.1 Ti rotor, 4 °C). The Anti-FLAG® M2-affinity gel (Merck) was washed twice and then resuspended in the binding buffer (purification buffer, 0.02% DDM). This was then added to the solubilised membrane and left to rotate for 1 hr at 4 °C. The resin was then pellet by centrifugation (300 g, 2 mins) and resuspended in the binding buffer and transferred to a spin column (Pierce). The resin was then washed with the wash buffer (purification buffer, 0.003% GDN) to exchange detergent systems (3 washes with 4x resin volume), by gravity. The complex was then eluted off with the elution buffer (buffer system, 0.003% GDN and 150 μg/mL 3X FLAG® peptide (Merck)), 1x resin volume by gravity, 3 elutions. The elution was then concentrated with 300 kDa concentrators to remove FLAG peptide and to concentrate the protein sample required concentration.

Spectra of the purified supercomplex were collected using a cuvette system with a similar cytochrome spectroscopy setup to the bioenergetic chamber except the quartz crucible is replaced with a cuvette, allowing transmission measurements (10 mm pathlength); the volume of the cuvette is 2.1 mL. The difference spectrum was collected by subtracting the reduced spectrum (sodium dithionite) from the oxidised spectrum (K₃[Fe(CN)₆]).

### ΔΨ and ΔpH estimates on *M. smegmatis*

ΔΨ and ΔpH was measured as in Rao et al with some adjustments[72]. Cells were prepared as for the chamber, pelleted and washed in media without Tween-80 and resuspended to an $OD_{600}$ of 1. The cells were then split and incubated at 37 °C in the 5 different conditions for 1 min: cells alone, DMSO, bedaquiline (30 nM) and DCCD (10 mM)). This was followed by the addition of [³H]TPP⁺ (5 μM) and a further incubation for 1 min at 37 °C. Next, cultures were centrifuged through 0.35 mL of silicon oil at 13000 × $g$ for 5 mins, 22 °C. 20 μL of supernatant samples were removed and the tube and rest of the contents were frozen at −80 °C overnight. The frozen pellet was removed with dog nail clippers. The supernatant and the pellet were mixed with 2 mL of scintillation fluid and placed in a scintillator to measure radioactivity. ΔΨ was estimated based on the uptake of [³H]TPP⁺ according to the Nernst equation. Non-specific binding of TPP⁺ was estimated from the cells that had been treated with valinomycin.

Estimates for ZΔpH were performed side by side with ΔΨ, except [7-¹⁴C]benzoate (5 μM, pKa 4.2) was added rather than [³H]TPP⁺ and valinomycin replaced nigericin (15 nM). ΔpH was estimated by the distribution of [7-¹⁴C]benzoate using the Henderson-Hasselbalch equation. ZΔpH was calculated as $\frac{-2.3RT}{F} \times 1000 \times \Delta pH$ which ~ $62 \times \Delta pH$. For comparisons between conditions, a repeated-measures ANOVA was conducted using JASP Version 0.19.0 (JASP Team) with a Bonferroni post-hoc test.

### Reporting summary

Further information on research design is available in the Nature Portfolio Reporting Summary linked to this article.

## Data availability

A source data file is available for this work. Source data are provided with this paper.

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

## Acknowledgements

We thank Dr Pooja Gupta (University of York), Dr Kiel Hards (University of Otago), and Dr Aneurin Kennerly (Manchester Metropolitan University) for formative discussions and Dr Matthew McNeil (University of Otago) for help with strains and microbiology. YSBL technicians and the Department of Chemistry and Biology workshops (University of York) provided critical support. This work was funded by a UKRI Future Leader Fellowship to JNB (MR/T040742/1), which directly supported MMO, and the Health Research Council of New Zealand. Work in JNB's laboratory is also supported by a BBSRC sLoLa award (BB/X003035/1). SHH was supported by a Department of Chemistry teaching PhD studentship and RCW with a BBSRC White Rose PhD Studentship (2434192); RCW was also supported by a travel grant funded by a philanthropic award from Dr Tony Wild (Wild Visiting Scholars Fund).

## Author contributions

Author contributions are reported below in line with the Contributor Roles Taxonomy (CRediT). Conceptualisation: S.H.H., R.C.W., G.M.C., M.M.O., J.N.B. Methodology: S.H.H., R.C.W., C.Y.C., R.J.S. G.M.C., M.M.O., J.N.B. Software: R.C.W., R.J.S., M.M.O. Validation: S.H.H., R.C.W., R.J.S., M.M.O., J.N.B. Formal analysis: R.C.W., R.J.S., M.M.O., J.N.B. Investigation: S.H.H., R.C.W., C.Y.C., M.M.O. Resources: R.J.S., G.M.C., M.M.O. Data Curation: S.H.H., R.C.W., M.M.O., J.N.B. Writing - original draft: S.H.H., R.C.W., M.M.O., J.N.B. Writing—review and editing: S.H.H., R.C.W., C.Y.C., R.J.S., G.M.C., M.M.O., J.N.B. Visualisation: S.H.H., R.C.W., M.M.O., J.N.B. Supervision: G.M.C., M.M.O., J.N.B.

## Competing interests

RJS is the founder of Cellspex Ltd, a company working to commercialise the remission spectroscopy technology used in this work. The remaining authors declare no competing interests.
