## [Transparent Peer Review file · Nature Communications]

Remission spectroscopy resolves the mechanism of action of bedaquiline within living mycobacteria

Corresponding Author: Professor James Blaza

Version 0:

Reviewer comments:

Reviewer #1

(Remarks to the Author)

What are the noteworthy results?

Licencing of Bedaquiline has transformed treatment of drug resistant TB identifying ATP-synthase, the electron transport chain and the respiratory chain as important targets for the development of novel antibiotics. However understanding its mechanism of action is critical to develop the next bedaquilines. A controversy in the field has been whether it acts as an uncoupler and/or an atypical ionophore and that this explains the rather contradictory increase in oxygen consumption rate on addition of bedaquiline.

This eloquently written manuscript not only resolves this controversy demonstrating that bedaquiline does not function as an protonophoric or ionophoric uncoupler demonstrating that its primary mode of action is through the direction inhibition of ATP synthase. This work identifies cytochrome bd oxidase role in bedquiline tolerance through relieving bedaquiline induced back-pressure. Before this work backpressure wasn't really understood in mycobacteria.

Will the work be of significance to the field and related fields? How does it compare to the established literature? If the work is not original, please provide relevant references.

This work is original and will be of interest to a wide range of audiences. In addition to the important biological findings with regard to how bedaquiline acts on mycobacteria it will be of interest to those biologists studying bioenergetics in other prokaryotic and eukaryotic systems in addition to those developing antibiotics that target bacterial energetics. This work presents an approach for measuring bioenergetic in living bacterial and eukaryotic cells and there will also be of interest to the biophysicists.

Does the work support the conclusions and claims, or is additional evidence needed?

The data supports the conclusions

Are there any flaws in the data analysis, interpretation and conclusions? Do these prohibit publication or require revision?

No the experimental studies are well executed with suitable controls

Is the methodology sound? Does the work meet the expected standards in your field?

Yes

Is there enough detail provided in the methods for the work to be reproduced?

Yes there are enough details in the methodology for the work to be reproduced.

I have some minor points for the manuscript

L54-55. Im not sure what the author means when they say that "Critically, remission spectroscopy allows the use of turbid media so that measurements can be made on bacterial suspension cultures growing in defined conditions, offering a system for measuring mycobacterial bioenergetics and drug action within living cells." What do they mean by turbid media? Do they mean bacteria at high cell density? I would fragment and clarify this sentence. To me the novelty is in the living cells.

L85. *Mycobacterium smegmatis* needs to be in italics

The authors have used the model organism *Mycobacterium smegmatis* and the Class two *Mycobacterium tuberculosis* H37Rv Δ RD1 Δ panCD (mc26230). There are obvious reasons why this is the case and needs to be explained in a few sentences within the manuscript.

Bedaquiline was discovered through studies in *M. smegmatis* but there are some important differences here and there should be a few sentences to explain the advantages and disadvantages of using *M. smegmatis* in this study.

Secondly it needs to be explained clearly in the text that its not virulent Mtb that is being used for these experiments as its slightly misleading. There needs to be a few sentences explaining this strain. It is a metabolic mutant strain that is auxotrophic for pantothenate and lacks the critical RD1 region that is essential for Mtb to cause TB. Throughout the authors must refer to the strain as H37Rv Δ RD1 Δ panCD (mc26230) not Mtb as its an avirulent mutant strain.

Does this mutant strain of Mtb behave the same way as its virulent relative in terms of bedaquiline. There needs to be some discussion of this.

The below statement needs to be changed as H37Rv Δ RD1 Δ panCD (mc26230) is not a pathogen and replicates poorly in animal models and cannot cause TB.

"Mtb exposed to bedaquiline behaved similarly (Fig. 2C and 2F), demonstrating the effect is conserved in the pathogen."

Reviewer #2

(Remarks to the Author)

Harrison and colleagues present the description of a method to measure bioenergetics parameters in live bacteria and successfully applied it to (finally) show how bedaquiline disrupts oxidative phosphorylation in mycobacteria. This is a major leap forward in the field of bacterial physiology, as it breaks away from the models currently utilized, and it will open the possibility to answer many unsolved questions in the field. The methods are described in detail. Conclusions are well-measured and largely corroborated by the presented data.

To strengthen the manuscript, some points could be clarified or discussed in more detail:

- 1- The measurements are made in whole cells, which is certainly a major advantage of the approach. However, the measured signal is a composite of all the enzymes that bind cytochromes. Have the authors checked if Mtb proteome includes enzymes with the cytochromes at study beyond respiration enzymes, which contribute to the signal? A cytochrome bc-aa3 oxidase knockout in *M. smegmatis* (PMID: 33343552) could be a good control for the background signal.
- 2- What was the rationale for the inclusion of attenuated Mtb in just a few experiments? Why not using it in all the experiments?
- 3- If KCN inhibits cytochrome bc-aa3 oxidase, why did it not lead to an increase in OCR via cytochrome bd oxidase?
- 4- In line 191 the authors state "However, this effect is less pronounced in Mtb Δ cydAB". I would say, in this case, it is the opposite effect.
- 5- In Mtb, treatment with bedaquiline in the millimolar range led to a very limited decrease on OCR, when compared to Msm, down to the wt levels at pre-treatment state. It is not clear how increasing bedaquiline concentration can distinguish between the two hypothesis 1) limited inhibition of ATP synthase and 2) presence of additional backpressure relief mechanisms. Can't the profiles still be compatible with, for example, Mtb cytochrome bc-aa3 oxidase being more resistant to backpressure than the Msm enzyme? Have the ATP synthase inhibition kinetics by bedaquiline in Msm and Mtb ever been directly compared? This could help disambiguate the proposed hypotheses.
- 6- Inside mammalian cells bedaquiline is trafficked via lipid droplets (PMID: 31249058), thus it is possible that different cell lines will have different lipid droplet dynamics, making the access to mitochondria heterogeneous. Thus, I don't think the authors can draw such a broad ranging conclusion as "bedaquiline simply does not reach mammalian ATP synthase at sufficient concentration to cause inhibition"
- 7- In the same topic, the authors state in the discussion "Our Δ cydAB experiments in both Mtb and Msm show that any induction of an adaptive response beyond redirection of flux must either be absent or have a minimal effect on PMF". I think the limited effect of bedaquiline in Mtb, even at a millimolar concentration is not compatible with this conclusion. In Mtb, it is possible that additional mechanisms are present beyond the flux redirection to cyt bd oxidase.

Text formatting issues:

- 1- The legend in Fig.1 does not match what is depicted.

Reviewer #3

(Remarks to the Author)

The paper «Remission spectroscopy resolves the mode of action of bedaquiline within living mycobacteria» from Harrison et al addresses an interesting and pressing problem, i.e. how do drugs like BDQ impact the life of a pathogen. BDQ is known to rather specifically bind to the ATP synthase of *M. tuberculosis* and is in the meantime an approved drug, however showing relatively easily developing resistant strains, and a good understanding of its mode of action is relevant for developing better drugs for patients. Since its first description in the early 2000s, a number of papers have addressed its mode of action, using a variety of systems including purified proteins in liposomes as well as studies in intact cells. Recently, works like Lamprecht et al (Nature Comm) have used oxygen consumption measurements to investigate rerouting of electrons in the respiratory chain of mycobacteria. The work presented here takes the experimental system one step further and also includes

measurements of spectroscopic characteristics of the heme proteins involved. In short, if the hemes are getting reduced, it indicates a backpressure, i.e. electrons cannot be freely donated to oxygen. They made use of a system called remission spectroscopy that is able to measure visible light spectra in turbid suspensions, allowing them to measure in growing cultures. In parallel, they controlled oxygen consumption rates, allowing them to correlate these important parameters. Here, they used it to investigate the mode of action of BDQ in either *M. smegmatis* (Msm) or *M. tuberculosis* (Mtb).

This is an interesting and relevant manuscript that contains a large body of new data. The authors are careful not to overinterpret their data, and the paper is generally well written, although a bit short at times, where a bit more information would be required.

Not all data are straightforward to explain, and the authors are careful in not drawing too many conclusions. However, at some instances, inconsistencies are not discussed in enough detail.

Interestingly, in quite some instances, the two mycobacterial organisms behave differently in response to the drug. In my opinion, these differences are not enough discussed, as pointed out below. In addition, some of the findings seem also inconsistent with earlier findings from Lamprecht et al, e.g. the absence of Δ cyd effect after addition of BDQ. If this paper wants to stand side-by-side with the publication of Lamprecht et al (also published in Nature Comm), it must tackle the differences in the discussion part.

Major comments:

- Line 87: In the control experiment, KCN was used to block the bccaa3 complex and a drop in OCR was observed, which made sense at that point. Later, we read that if ATP synthase is blocked by BDQ, the OCR increases because the cell switches to bd oxidase instead. Why is this not happening after addition of KCN (which is not supposed to block bd oxidase, I think)? If bd oxidase is still available, and the ATP synthase functioning, we would expect an increase in OCR. Please clarify?
- Line 115/116: Again, the control experiments show that cytochrome a and c are most easily reduced, but not heme b (Figure 1E). Why is this not happening here with BDQ, while cyt b and c are reduced? This is really not easy to understand. Obviously, this can be explained in mitochondria (ref. 29). Please clarify!
- Line 142–145: There is quite a visible difference between the spectral shift behaviours of Msm and Mtb upon addition of CCCP. Is there anything to rationalize in terms of metabolic pathways that are expressed under these conditions?
- Line 149–154: A very large difference is also seen in the two organisms upon addition of nigericin. While this is mentioned, I would have hoped for a more elaborate explanation. What kind of event could explain the observed spectral changes?
- Line 176–177: The increase in OCR is not really impressive (Figure 1A). It seems much more consistent in Mtb, also seen in reference 14 (Lamprecht). There, only experiments with Mtb were made, including the Δ cyd knockout of bd oxidase. However, there, no impact on OCR was observed (still stimulated), while the authors found a pronounced impact here (decrease for Msm, slower OCR increase in Mtb). A more detailed comparison of these two studies with similar aims seems necessary.
- Line 196–198: There is a drastic difference in the behaviour of Mtb to either 120 nM or 10 μ M BDQ (slow and continuous vs. immediate and maxing out). On the other hand, little difference is seen in Msm upon addition of the two different concentrations. Is there any rationale behind that?
- Line 241: What do the authors mean by activated? Since the change is so rapid, de novo translation is unlikely, meaning that bd oxidase was there in the first place. Why is it not used then in competition with the supercomplex? Thermodynamically, the reaction in bd oxidase is more favorable than in the supercomplex. Do the authors envision an inhibition of bd oxidase? Obviously, this finding is also in contrast to what has been found in ref. 14, where a Δ cyd knockout had no effect on OCR. Instead, they found that inhibition of ATP synthase (BDQ or DCCD) both show increased OCR that can be mediated either via bd oxidase or bc1aa3. This discrepancy should be discussed in more detail.
- Line 265–266: This is an interesting observation and hypothesis, but needs some more clarification. Since Ndh-2 is a NADH:quinone oxidoreductase, backpressure means that the quinol pool is reduced and NDH-2 is looking for alternative substrates like clofazimine. However, we see an increase in bd activity and an overall more rapid O₂ consumption, suggesting that more quinol is required to reduce oxygen. Would the authors argue that BDQ shifts the Q-pool to the reduced side?

Minor comments:

- The term attenuation should be briefly introduced.
- Something is off with the labeling in Figure 1 and in the text. While the figure shows only panels A to D, panels up to H appear in the legend and in the text.
- How was Figure 1E or H (the purified supercomplex) created? Was this also measured in the 5 ml chamber, explaining the rasterized spectrum? What concentration was used?
- mOD: Does this stand for OD divided by 1000?
- One should write out the actual concentrations of CCCP, BDQ, and nigericin in the text. This is important given the fact that the outcomes are different. Using terms like low and higher seems not appropriate.
- Figure 2A: What is the reason for the temporary drop in OCR at 1.5 μ M BDQ?
- Figure 3G: What is the signal around 580 nm? It has not been addressed.
- I didn't find an explanation in the text, why 30 and 120 nM BDQ are used for Msm and Mtb.
- Line 162–163: Why is a more dense culture required for the Mtb compared to the Msm?
- Line 171–172: It is acting like an oligomycin-like inhibitor, except for the lack of OCR decrease. OCR decrease is a cornerstone of oligomycin activity. I know it is mentioned in the next paragraph, but standing alone, the statement is not correct.

- Line 178–179: The statement is only partially correct. Next to electroneutral proton release to the periplasm, there is an electrogenic uptake of protons to the catalytic site in bd oxidases which contributes to the $\Delta\psi$ and thus the pmf.
- Line 512: 10 mM DCCD is high. Which concentration was used?

Version 1:

Reviewer comments:

Reviewer #2

(Remarks to the Author)

The authors satisfactorily addressed all my questions.

Reviewer #3

(Remarks to the Author)

Harrison et al have done a very thorough job addressing all the comments of the different reviewers. The revised version of the manuscript is now significantly longer and addresses critical points. The authors acknowledge that not all data can be perfectly explained but they provide possible explanations and testable ideas. And I hope they will test some of these explanations in the future. The methods applied here combining visible light spectroscopy and OCR measurements is a pioneering work and reflects well the complexity of bioenergetics in general or cellular redox balance in particular. I am still puzzled by some of the results they get (e.g. non-reduced a type hemes, different OCR simulations, nigericin results), but I acknowledge that they have done these experiments as thoroughly as possible and they deserve publications in these form. As pointed out by the authors, future studies, including such with purified membranes or proteins will test their ideas on how to activate bd oxidases within seconds or if yet another redox balance has to be considered. Please find below some small minor comments that occurred to be during reading.

P3, line 85. According to the internet, saprothropic should be used instead of saprophytic, if the organism in question is not a plant.

P7, line 217. Instead of writing generic Figure 2, the exact panels should be specified. Here 2A, D. If other unprecise instances exist, they should also be corrected.

P7, line 221-223. The interpretation with the off-target effects at 25 μM CCCP is possibly correct, but the amount of data shown here might not substantially support this interpretation.

P7, line 233. I don't understand how nigericin builds a $\Delta\psi$, if doing electroneutral K/H (1:1) exchange. This is different behavior than valinomycin, which follows Nernst-potential behavior.

P7, 235-6. The nigericin behavior is a riddle to me. A large drop in OCR means a higher backpressure, resulting in higher cytochrome occupancy, but the opposite is seen. Then after a few minutes the whole recovers. But the authors do a good job describing their observations.

Page 10, line 326: great should be greatly

P12, line 375-77. This sounds a bit vague, and could be formulated more clearly.

Response to reviews for ‘Remission spectroscopy resolves the mode of action of bedaquiline within living mycobacteria’

Suzanna H Harrison^{1,2*}, Rowan C Walters^{1,2*}, Chen-Yi Cheung³, Roger J Springett^{1,2,4}, Gregory M Cook^{3,5}, Morwan M Osman^{1,2†*}, James N Blaza^{1,2†}

¹York Structural Biology Laboratory, Department of Chemistry, University of York, York, YO10 5DD. ²York Biomedical Research Institute, University of York, York, YO10 5DD. ³Department of Microbiology and Immunology, University of Otago, Dunedin 9016, New Zealand. ⁴Cellspex Ltd, Northamptonshire, UK. ⁵ School of Biomedical Sciences, Queensland University of Technology, Brisbane, Queensland 4000, Australia.

0.1. We thank the reviewers for their supportive comments and for taking the time to write thoughtful and critical observations that we are sure will make the manuscript stronger and more accessible. Before addressing the reviewer comments we have some overarching things to mention.

0.2. We have realised there was a bug in Google Docs that led to Fig 1 being an earlier version when an image is replaced with an other (rather than being deleted and then re-inserted). This meant when we used File > save as pdf to create the pdf for submission an old image was inserted. The preprint did not have this problem because it was downloaded as a Word document beforehand and so the image didn't revert. We are sincerely sorry for this oversight and apologise to the reviewers. The correct figure is now in and double checked.

0.3. The reviewers all focused on different parts of the work and appeared to be from quite different backgrounds so there were no overriding ‘themes’ to address, although a few individual points were raised more than once. Addressing reviewers comments has meant the manuscript is now much longer, just under the 5k word limit allowed for Nat Comms.

0.4. We realised that some of the Results and Figures could be split to make the paper more accessible and better structured. We also realised that we could expand upon the KCN data in Fig 1 to make the paper more complete. The new KCN data makes a new Fig. 5 and the human data has been split out of Fig. 4 to make a new Fig. 6. In both cases, a new subheading has been introduced to the Results section: ‘Slowing the supercomplex directly induces electron flux redirection to CydAB’ and ‘Human mitochondria are unaffected by bedaquiline’.

0.5. We have rewritten the abstract to be more informative and accessible based on what landed with the reviewers and comments that people have made since the preprint was published. It now reads:

Bedaquiline, an ATP synthase inhibitor, is the spearhead of transformative therapies against drug-resistant *Mycobacterium tuberculosis*. Here, remission spectroscopy is used to measure the energy-transducing cytochromes within unperturbed, respiring suspensions of mycobacterial and human cells, allowing spectroscopic measurements of electron transport chains as they power living cells and respond to bedaquiline. No evidence is found for protonophoric or ionophoric uncoupling. Rather, by directly inhibiting ATP synthase, bedaquiline slows the respiratory supercomplex (Qcr:Cta; *bcc:aa₃*) by increasing the proton-motive force, causing sub-second redirection of electron flux through the cytochrome *bd* oxidase (Cyd) to O₂. Electron flux redirection explains the idiosyncratic bedaquiline-induced increase in O₂ consumption rates previously observed. Redirection occurs as Cyd is present even in cells grown in plentiful O₂. Applying the same approach to human cells did not detect bedaquiline-induced inhibition of mitochondrial function despite such inhibition being seen in isolated systems. Overall, we clarify how bedaquiline works, why different models for its action developed, and the mechanisms underlying the synergy of bedaquiline in combination regimes.

0.6. Line numbers refer to the new document as it is much changed.

0.7. We have included source data in an .xlsx file, which can be hosted with the paper if published.

Reviewer #1 (Remarks to the Author):

What are the noteworthy results?

Licensing of Bedaquiline has transformed treatment of drug resistant TB identifying ATP-synthase, the electron transport chain and the respiratory chain as important targets for the development of novel antibiotics. However understanding its mechanism of action is critical to develop the next bedaquilines. A controversy in the field has been whether it acts as an uncoupler and/or an atypical ionophore and that this explains the rather contradictory increase in oxygen consumption rate on addition of bedaquiline.

This eloquently written manuscript not only resolves this controversy demonstrating that bedaquiline does not function as a protonophoric or ionophoric uncoupler demonstrating that its primary mode of action is through the direct inhibition of ATP synthase. This work identifies cytochrome *bd* oxidase role in bedaquiline tolerance through relieving bedaquiline induced back-pressure. Before this work backpressure wasn't really understood in mycobacteria.

Will the work be of significance to the field and related fields? How does it compare to the established literature? If the work is not original, please provide relevant references.

This work is original and will be of interest to a wide range of audiences. In addition to the important biological findings with regard to how bedaquiline acts on mycobacteria it will be of interest to those biologists studying bioenergetics in other prokaryotic and eukaryotic systems in addition to those developing antibiotics that target bacterial energetics. This work presents an approach for measuring bioenergetic in living bacterial and eukaryotic cells and there will also be of interest to the biophysicists.

Response to reviewer 1.1: We thank the reviewer for their very kind remarks.

Does the work support the conclusions and claims, or is additional evidence needed?

The data supports the conclusions

Are there any flaws in the data analysis, interpretation and conclusions? Do these prohibit publication or require revision?

No the experimental studies are well executed with suitable controls

Is the methodology sound? Does the work meet the expected standards in your field?

Yes

Is there enough detail provided in the methods for the work to be reproduced?

Yes there are enough details in the methodology for the work to be reproduced.

I have some minor points for the manuscript

L54-55. I'm not sure what the author means when they say that "Critically, remission spectroscopy allows the use of turbid media so that measurements can be made on bacterial suspension cultures growing in defined conditions, offering a system for measuring mycobacterial bioenergetics and drug action within living cells." What do they mean by turbid media? Do they mean bacteria at high cell density? I would fragment and clarify this sentence. To me the novelty is in the living cells.

1.2. Yes, we mean bacteria growing at a high cell densities. We now have a paragraph that hopefully better explains our approach (line 100-119):

'In this work, remission spectroscopy is used to measure cytochromes in living cells. Cytochromes (*cyto-*, cellular and *chrome*, colour) are haem-containing redox enzymes. Many of the ETC complexes are cytochromes as their haems act as integral mechanistic redox centres. Absorbing light strongly, and that absorbance depending on the oxidation state of the Fe atom, renders cytochromes excellent spectroscopic handles for electron transfer. Originally observed in intact organisms,²⁵ the strong scattering of visible-wavelength light by cells means detailed spectroscopic measurement of

cytochromes has historically relied on fractionating cells to subcellular structures such as mitochondria^{26,27} or isolated enzymes.²⁸ However, for intact cells, remission spectroscopy provides a method to measure spectra in highly-scattering cell suspensions.²⁹ Here, a stable light source illuminates the culture through a small aperture and the light that has been scattered 180° ('re-emitted') through another small aperture is detected. Because of the set-up, remission spectroscopy cannot provide absolute absorbance spectra relative to a spectroscopically clear reference but instead provides deflections from a baseline ('difference spectra'). Remission-spectroscopic measurements made on cell suspensions have a differential pathlength that varies less than 3% from ~530 nm to 740 nm,³⁰ which provides sufficiently accurate remission spectra that known spectra of isolated cytochromes can be additively combined to recreate the observed signal, attesting to the spectral fidelity of the approach.³¹ A note on terminology: formally, all visible-wavelength spectroscopies measure attenuation, the depletion of light reaching the detector by both scattering or absorption of photons by an analyte, but over time absorbance has become the accepted, if incorrect, term. Here, we use attenuation and absorbance in their formal senses.'

L85. *Mycobacterium smegmatis* needs to be in italics

1.3. Fixed. Thank you.

The authors have used the model organism *Mycobacterium smegmatis* and the Class two *Mycobacterium tuberculosis* H37Rv Δ RD1 Δ panCD (mc26230). There are obvious reasons why this is the case and needs to be explained in a few sentences within the manuscript.

Bedaquiline was discovered through studies in *M. smegmatis* but there are some important differences here and there should be a few sentences to explain the advantages and disadvantages of using *M. smegmatis* in this study.

Secondly it needs to be explained clearly in the text that its not virulent Mtb that is being used for these experiments as its slightly misleading. There needs to be a few sentences explaining this strain. It is a metabolic mutant strain that is auxotrophic for pantothenate and lacks the critical RD1 region that is essential for Mtb to cause TB. Throughout the authors must refer to the strain as H37Rv Δ RD1 Δ panCD (mc26230) not Mtb as its an avirulent mutant strain.

Does this mutant strain of Mtb behave the same way as its virulent relative in terms of bedaquiline. There needs to be some discussion of this.

1.4. Agreed. The full name is rather long through, so we now use *M. tuberculosis* mc²6230 in all points in the Results section to make it clear and have introduced the following paragraph in the introduction, which we hope satisfies the reviewer (line 82-99):

'*M. tuberculosis* is a demanding organism to work with. It grows slowly, has a fastidious nature, and readily spreads through aerosols, making it a level 3 biosafety agent. For these reasons, *Mycobacterium smegmatis* has found wide use as a model mycobacterium for molecular physiology. A saprophytic microbe that grows relatively quickly, for which a range of powerful genetic tools have been developed, has led to *M. smegmatis* being described as the 'vanguard of mycobacterial research'.¹⁸ Supporting the utility of *M. smegmatis* for bioenergetic research, bedaquiline was discovered with this organism,⁶ and the respiratory complexes are highly conserved, as shown in high-resolution structures determined from both species such as ATP synthase^{9,10} and the *bd* oxidase (CydAB).^{19,20} *M. smegmatis* has a growing role as a bioenergetic model organism, for example, providing insights into aerobic H₂ and CO oxidation.^{21,22} However, the physiology of an organism that divides every 4 hours will be distinct from one doubling every ~24 hours (or longer). For work closer to *M. tuberculosis* that is unsuitable for level 3 biosafety conditions, *M. tuberculosis* strains mutated to be less pathogenic, typically developed for vaccines, offer a suitable alternative. Bacillus Calmette–Guérin (BCG) strains are much less pathogenic but have accumulated extensive genomic changes over decades of culture.²³ To address this, a commonly used *M. tuberculosis* pathogenic cell

line (H37Rv) has had the main inactivating mutation from BCG introduced ($\Delta RD1$) alongside pantothenate autotrophy ($\Delta panCD$) to give rise to mc²6230 or mc²6030.²⁴ *M. tuberculosis* mc²6230 is therefore less infectious and carries a metabolic defect and is used here.'

The below statement needs to be changed as H37Rv $\Delta RD1 \Delta panCD$ (mc26230) is not a pathogen and replicates poorly in animal models and cannot cause TB.

"Mtb exposed to bedaquiline behaved similarly (Fig. 2C and 2F), demonstrating the effect is conserved in the pathogen."

1.5 Agreed. This now reads, '*M. tuberculosis* mc²6230 exposed to 120 nM bedaquiline behaved similarly (Fig. 2C and 2F), demonstrating the effect is conserved in both mycobacterial species.' (line 181 & 182)

Reviewer #2 (Remarks to the Author):

Harrison and colleagues present the description of a method to measure bioenergetics parameters in live bacteria and successfully applied it to (finally) show how bedaquiline disrupts oxidative phosphorylation in mycobacteria. This is a major leap forward in the field of bacterial physiology, as it breaks away from the models currently utilized, and it will open the possibility to answer many unsolved questions in the field. The methods are described in detail. Conclusions are well-measured and largely corroborated by the presented data.

2.1 We thank the reviewer for their positive assessment of our work.

To strengthen the manuscript, some points could be clarified or discussed in more detail:

1- The measurements are made in whole cells, which is certainly a major advantage of the approach. However, the measured signal is a composite of all the enzymes that bind cytochromes. Have the authors checked if Mtb proteome includes enzymes with the cytochromes at study beyond respiration enzymes, which contribute to the signal? A cytochrome bc-aa3 oxidase knockout in *M. smegmatis* (PMID: 33343552) could be a good control for the background signal.

2.2. All our measurements are made as a relative change from baseline, where the baseline is the cells respiring in the chamber. While other signals will certainly be present, we only measure signals that change and so these are not a problem. We have put a paragraph into the introduction to help clarify the spectroscopy; see 1.2 above.

2- What was the rationale for the inclusion of attenuated Mtb in just a few experiments? Why not using it in all the experiments?

2.3 We think that the mycobacteria genus as a whole is interesting and indeed the findings on the pressure relief activity of Cyd is applicable to the whole genus. The introduction now better explains this, reflecting recent results on *M. smegmatis* (line 83-91):

'For these reasons, *Mycobacterium smegmatis* has found wide use as a model mycobacterium for molecular physiology. A saprophytic microbe that grows relatively quickly, for which a range of powerful genetic tools have been developed, has led to *M. smegmatis* being described as the 'vanguard of mycobacterial research'.¹⁸ Supporting the utility of *M. smegmatis* for bioenergetic research, bedaquiline was discovered with this organism,⁶ and the respiratory complexes are highly conserved, as shown in high-resolution structures determined from both species such as ATP synthase^{9,10} and the bd oxidase (CydAB).^{19,20} *M. smegmatis* has a growing role as a bioenergetic model organism, for example, providing insights into aerobic H₂ and CO oxidation.^{21,22}

3- If KCN inhibits cytochrome bc-aa3 oxidase, why did it not lead to an increase in OCR via cytochrome bd oxidase?

2.4 This is a really interesting question. Indeed, KCN does cause an increase in OCR at low concentrations. We have introduced a new figure, Fig. 5, to address this point, which is structured in

its own section, 'Slowing the supercomplex directly induces electron flux redirection to CydAB' (lines 310-326).

4- In line 191 the authors state "However, this effect is less pronounced in *Mtb* Δ cydAB". I would say, in this case, it is the opposite effect.

2.5. Agreed. We have changed the wording to read:

'However in *M. tuberculosis* Δ cydAB OCR still increases, but much less than in *M. tuberculosis* mc²6230 (Fig 4B).'

5- In *Mtb*, treatment with bedaquiline in the millimolar range led to a very limited decrease on OCR, when compared to *Msm*, down to the wt levels at pre-treatment state. It is not clear how increasing bedaquiline concentration can distinguish between the two hypothesis 1) limited inhibition of ATP synthase and 2) presence of additional backpressure relief mechanisms. Can't the profiles still be compatible with, for example, *Mtb* cytochrome bc-aa₃ oxidase being more resistant to backpressure than the *Msm* enzyme? Have the ATP synthase inhibition kinetics by bedaquiline in *Msm* and *Mtb* ever been directly compared? This could help disambiguate the proposed hypotheses.

2.6. This is a very interesting and relevant point.

Firstly, a clarification: we never work at mM concentrations bedaquiline, the highest concentration we work at is 10 μ M.

If the *Mtb* supercomplex is more resistant to backpressure or there are other compensatory mechanisms then the concentration of bedaquiline will not matter as the adaptive systems will engage and OCR will continue as normal. If bedaquiline influx or efflux causes the differences, then increasing [bedaquiline] should lead to more bedaquiline getting into cells based on first order diffusion of bedaquiline into the cells. The paragraph has been tweaked to read:

'When *M. smegmatis* Δ cydAB was exposed to 30 nM bedaquiline there was an immediate decrease in OCR, whilst the electron occupancy of *b* and *c* cytochromes increased as before (Fig 4A&E). This behaviour is similar with the behaviour of oligomycin acting on mitochondria, demonstrating that CydAB mediates the increase in OCR in response to bedaquiline. However, in *M. tuberculosis* Δ cydAB OCR still increased, but much less than in *M. tuberculosis* mc²6230 (Fig 4B). We speculated this was due to one of two reasons: (1) bedaquiline is less capable of reaching its target site in *M. tuberculosis* mc²6230 through differences in bedaquiline influx or efflux or (2) the bioenergetic system of *M. tuberculosis* has additional mechanisms to compensate for the backpressure induced by bedaquiline compared to *M. smegmatis*. In support of (1), we noticed that tuberculosis responded much more slowly to bedaquiline than to other small molecule effectors like nigericin and CCCP (Fig 2C and Fig 3B&D). If the differences were due to the ability of bedaquiline to reach its target site, increasing the concentration of bedaquiline should overcome it. When the concentration of bedaquiline was increased to 10 μ M, there was a robust decrease in OCR in both *M. smegmatis* and *M. tuberculosis* Δ cydAB (Fig 4C,D). The differences observed may have been due to the differences in expression of the MmpS5-MmpL5 transporter, which is implicated in bedaquiline efflux.⁴⁹

While comparative bedaquiline inhibition kinetics have not been studied in detail to our knowledge, the lack of rapid bedaquiline efflux from *Msm* was probably why bedaquiline was discovered in this organism in a high throughput screen.

6- Inside mammalian cells bedaquiline is trafficked via lipid droplets (PMID: 31249058), thus it is possible that different cell lines will have different lipid droplet dynamics, making the access to mitochondria heterogeneous. Thus, I don't think the authors can draw such a broad ranging conclusion as "bedaquiline simply does not reach mammalian ATP synthase at sufficient concentration to cause inhibition"

2.7. Our conclusion was reached by citing the reference the reviewer has raised but we agree the original wording was overly strong. This now reads as (line 346-351):

Therefore, we observed no evidence for bedaquiline inhibiting mammalian cells whilst inducing adaptive mechanisms analogous to those found in mycobacteria. Combined with observations that bedaquiline is found in mitochondria at low concentrations compared to other parts of the cell⁵⁴ we conclude that bedaquiline either does not enter mitochondria, or enters and is rapidly pumped out, and therefore leaves mammalian ATP synthase unaffected.

7- In the same topic, the authors state in the discussion “Our Δ cydAB experiments in both Mtb and Msm show that any induction of an adaptive response beyond redirection of flux must either be absent or have a minimal effect on PMF”. I think the limited effect of bedaquiline in Mtb, even at a millimolar concentration is not compatible with this conclusion. In Mtb, it is possible that additional mechanisms are present beyond the flux redirection to cyt bd oxidase.

2.8. We are unsure if the reviewer means limited effect in terms of what is measured here or limited effect in that when exposed to bedaquiline, Mtb survives for weeks.

The full paragraph now reads (line 365-372):

‘Previously, the nature of backpressure within mycobacterial bioenergetics had been questioned, with proposals that it is either not sensed or collapsed by energy spilling mechanisms.¹⁴ By directly measuring the status of the ETC *in vivo*, we find that bedaquiline increases backpressure and that the cytochromes of the ETC directly sense it. Our Δ cydAB experiments in both *M. tuberculosis* mc²6230 and *M. smegmatis* show that any induction of an energy spilling mechanism, beyond redirection of electron flux, must either be absent or have a minimal effect on PMF. This backpressure explains why bedaquiline induces increased succinate accumulation and elevated NADH/NAD⁺ levels,^{12–14} signatures of high PMF and backpressure in mammalian systems.^{54,55}’

This paragraph is primarily our response to the conclusion of Lamprect et al that, ‘Our data shows that, unlike in eukaryotes, back pressure does not substantially impede ETC activity; this suggests that an energy spilling pathway modulates PMF to prevent excessive accumulation’.

We have expanded the paragraph to specifically refer to energy spilling.

Text formatting issues:

1- The legend in Fig.1 does not match what is depicted.

2.9. Thank you for catching this. Please see 0.2 for our explanation and accept our apologies.

Reviewer #3 (Remarks to the Author):

The paper «Remission spectroscopy resolves the mode of action of bedaquiline within living mycobacteria» from Harrison et al addresses an interesting and pressing problem, i.e. how do drugs like BDQ impact the life of a pathogen. BDQ is known to rather specifically bind to the ATP synthase of *M. tuberculosis* and is in the meantime an approved drug, however showing relatively easily developing resistant strains, and a good understanding of its mode of action is relevant for developing better drugs for patients. Since its first description in the early 2000s, a number of papers have addressed its mode of action, using a variety of systems including purified proteins in liposomes as well as studies in intact cells. Recently, works like Lamprecht et al (Nature Comm) have used oxygen consumption measurements to investigate rerouting of electrons in the respiratory chain of mycobacteria. The work presented here takes the experimental system one step further and also includes measurements of spectroscopic characteristics of the heme proteins involved. In short, if the hemes are getting reduced, it indicates a backpressure, i.e. electrons cannot be freely donated to oxygen. They made use of a system called remission spectroscopy that is able to measure visible light spectra in turbid suspensions, allowing them to measure in growing cultures. In parallel, they controlled oxygen consumption rates, allowing them to correlate these important parameters.

Here, they used it to investigate the mode of action of BDQ in either *M. smegmatis* (Msm) or *M. tuberculosis* (Mtb).

This is an interesting and relevant manuscript that contains a large body of new data. The authors are careful not to overinterpret their data, and the paper is generally well written, although a bit short at times, where a bit more information would be required.

3.1 We thank the reviewer for their positive assessment. The paper is now much longer and hopefully does better justice both to the field(s) and our results.

Not all data are straightforward to explain, and the authors are careful in not drawing too many conclusions. However, at some instances, inconsistencies are not discussed in enough detail.

3.2 We agree, and trust the increased length of the manuscript will help here.

Interestingly, in quite some instances, the two mycobacterial organisms behave differently in response to the drug. In my opinion, these differences are not enough discussed, as pointed out below. In addition, some of the findings seem also inconsistent with earlier findings from Lamprecht et al, e.g. the absence of Δ cyd effect after addition of BDQ. If this paper wants to stand side-by-side with the publication of Lamprecht et al (also published in Nature Comm), it must tackle the differences in the discussion part.

3.3. These are good points and we respond to them below.

Major comments:

- Line 87: In the control experiment, KCN was used to block the *bccaa3* complex and a drop in OCR was observed, which made sense at that point. Later, we read that if ATP synthase is blocked by BDQ, the OCR increases because the cell switches to bd oxidase instead. Why is this not happening after addition of KCN (which is not supposed to block bd oxidase, I think)? If bd oxidase is still available, and the ATP synthase functioning, we would expect an increase in OCR. Please clarify?

3.4 Excellent point. We have done the experiment titrating in KCN and put the results as a new section and figure ('Slowing the supercomplex directly induces electron flux redirection to CydAB', line 317; Fig. 5).

- Line 115/116: Again, the control experiments show that cytochrome a and c are most easily reduced, but not heme b (Figure 1E). Why is this not happening here with BDQ, while cyt b and c are reduced? This is really not easy to understand. Obviously, this can be explained in mitochondria (ref. 29). Please clarify!

3.5 At the moment, we can only speculate, and understanding these changes properly will be the focus of upcoming papers. Probably it will keep us busy for some time.

Our reasoning is that in fig 1, we are inhibiting the active site of the *aa₃* oxidase (the binuclear centre; BNC) with CN⁻, which will cause electrons to accumulate from the 'a-haem end' back up, in line with thermodynamic predictions. However, bedaquiline causes PMF to increase which will affect any haem centre within the membrane dielectric and every proton pumping site in a complex way. The BNC is composed of haem *a₃* and a copper ion; haem *a* sits next to it. The BNC has a complex reaction cycle, composed of 4 electron transfers, 4 pumped 'vectorial' protons and 4 chemical protons that drive the complete reduction of O₂ to H₂O. Because *a₃* has a very flat spectrum in its most abundant reduced state, we are unable to measure it, but if its occupancy is changing then it may be that it stops haem *a* being reduced through electrostatic interactions.

- Line 142–145: There is quite a visible difference between the spectral shift behaviours of Msm and Mtb upon addition of CCCP. Is there anything to rationalize in terms of metabolic pathways that are expressed under these conditions?

3.6 Probably not metabolic pathways as everything feeds into the electron transport chain at Q. However, we think the pathways are run at different levels of electron occupancies and turnover numbers. We hope to measure this rigorously in the future. For now, we have added the following sentence: 'The differences between them probably reflect differences in the pre-treatment

electron-occupancies and turnover frequencies of the ETCs in *M. smegmatis* and *M. tuberculosis* mc²6230⁷.

- Line 149–154: A very large difference is also seen in the two organisms upon addition of nigericin. While this is mentioned, I would have hoped for a more elaborate explanation. What kind of event could explain the observed spectral changes?

3.7 We presume that there are differences in the ion balances that the two organisms sustain across their membranes. Having done a literature and bioinformatic we have had to conclude that there really is not much known about ion balance in mycobacteria, apart from a little on K⁺ homeostasis. We have expanded this paragraph to expand on the similarities and differences. Ultimately, while these results are interesting, and we do want to build upon them in the future in a way that is currently unclear, the signature of the changes is completely different to those induced by bedaquiline. Given the importance of and interest in bedaquiline, we prefer to keep speculation to a minimum.

- Line 176–177: The increase in OCR is not really impressive (Figure 1A). It seems much more consistent in Mtb, also seen in reference 14 (Lamprecht). There, only experiments with Mtb were made, including the Δ cyd knockout of bd oxidase. However, there, no impact on OCR was observed (still stimulated), while the authors found a pronounced impact here (decrease for Msm, slower OCR increase in Mtb). A more detailed comparison of these two studies with similar aims seems necessary.

3.8. Thank you for this observation. We have now included the below two paragraphs in the discussion covering this issues:

On whether CydAB was really deleted in the strains used by Lamprecht et al (line 414-422):

Our observations around O₂ consumption and bedaquiline are consistent with Lamprecht et al,¹⁴ with a single exception: that deletion of *cydAB* stops the bedaquiline-induced increase in OCR. That study used a strain of *M. tuberculosis* nominally deleted in *cyd*,⁷¹ but the strain was later revealed to only harbour a small deletion of *cydB*, all of *cydD*, and much of *cydC*, leaving *cydAB* mostly intact.⁷² CydDC shares its operon with CydAB but does not constitute the oxidase; in *E. coli* CydDC is a haem transporter and it is unclear if deleting CydDC affects CydAB function.⁷³ In this work, we used an independently generated deletion of the full CydAB open reading frames,^{45,46} avoiding this caveat. As an aside, we note that this observation is only possible is due to commendable decision of Arora et al to publish a clarification of their prior work.⁷²

On OCR changes (line 413-424):

Measurements of the bedaquiline-induced increase in OCR by other workers have found its magnitude to be larger than ours but this may be due to the inclusion of a carbon-source-free ‘starvation’ step before bedaquiline addition. For example, a suspension culture measurement of *M. smegmatis* respiring within a Clark electrode, similar in approach to our measurements here, displayed a ~100% increase in OCR but was made on cells that had been resuspended in carbon-free phosphate-buffered saline before glycerol was re-introduced and bedaquiline introduced.¹⁵ Measurements using a XF96 Extracellular Flux Analyser on a monolayer of *M. tuberculosis* H37Rv cells found an even greater increase: ~600%.¹⁴ These were made on cells that had been starved in carbon-free medium for 24 hours. Our approach relied on taking mid-exponential phase bacteria, washing them in detergent-free growth medium so they remain energised, and quickly placing them in the chamber. Therefore, they should remained in the mid-exponential growth phase when we conducted our experiments. We conclude the magnitude of the bedaquiline-induced increase in OCR is dependent on exact growth and measurement conditions.

- Line 196–198: There is a drastic difference in the behaviour of Mtb to either 120 nM or 10 μ M BDQ (slow and continuous vs. immediate and maxing out). On the other hand, little difference is seen in Msm upon addition of the two different concentrations. Is there any rationale behind that?

3.9. See point 2.6 to reviewer 2

- Line 241: What do the authors mean by activated? Since the change is so rapid, de novo translation is unlikely, meaning that bd oxidase was there in the first place. Why is it not used then in competition with the supercomplex? Thermodynamically, the reaction in bd oxidase is more favorable than in the supercomplex. Do the authors envision an inhibition of bd oxidase? Obviously, this finding is also in contrast to what has been found in ref. 14, where a Δ cyd knockout had no effect on OCR. Instead, they found that inhibition of ATP synthase (BDQ or DCCD) both show increased OCR that can be mediated either via bd oxidase or bc1aa3. This discrepancy should be discussed in more detail.

3.10. Excellent point. The following paragraph has been inserted in the discussion (lines 393-402):

‘We lack a verified and unambiguous mechanism for how CydAB is activated but there are hints in the literature. As CydAB does not pump protons, the overall ΔG over the enzyme will be strong (~ 900 mV, ~ 86 kJ mol⁻¹) and it must be under kinetic control. Any mechanism needs to occur on a second time scale. Possible options include a regulatory switch and allostery; the two are not mutually exclusive. A disulphide bond next to the periplasmic quinol binding-site was observed in structures of mycobacterial CydAB.^{19,20,63} Reducing agents that break the disulphide decrease activity *in vitro*, possibly creating a link to ROS regulation.⁶³ CydAB may also be tuned to react with MKH₂ so that when the Q-pool fills with electrons, MKH₂ binds to CydAB at an allosteric site and activates it. Recombinant CydAB can now be produced in sufficient quantities for detailed kinetic analysis,^{63,64} raising the possibility of further detailed investigations on the isolated enzyme.’

- Line 265–266: This is an interesting observation and hypothesis, but needs some more clarification. Since Ndh-2 is a NADH:quinone oxidoreductase, backpressure means that the quinol pool is reduced and NDH-2 is looking for alternative substrates like clofazimine. However, we see an increase in bd activity and an overall more rapid O₂ consumption, suggesting that more quinol is required to reduce oxygen. Would the authors argue that BDQ shifts the Q-pool to the reduced side?

3.11. Excellent question. We have now included the following paragraph to answer (line 371-382):

‘The menaquinone-pool is a central metabolic hub. With the bedaquiline-induced increase in electron occupancy occurring in the supercomplex occurring simultaneously with the activation of CydAB that empties the menaquinol pool of electrons, the status of quinone pool is ambiguous. However, the supercomplex associated b_{564} signal acts as an indicator. The E_m values of the haems that make up b_{564} , b_L and b_H , in the closely related supercomplex from *Corynebacterium glutamicum* are more negative than those from ubiquinol-reactive bc_1 enzymes to enable efficient electron transfer.⁵⁷ Therefore, the assumption commonly made for mammalian enzymes, that the b -haems of bc_1 -family enzymes is close to or at equilibrium with the Q-pool,^{22,58} is reasonable here and we tentatively conclude that the increase in electron occupancy seen in b_{564} in *M. tuberculosis* mc²6230 and *M. smegmatis* means the menaquinone pool is filling with electrons overall.’

Minor comments:

- The term attenuance should be briefly introduced.

3.12. Good point. We have defined our terms in the spectroscopy introduction paragraph (line 116-119):

‘A note on terminology: formally, all visible-wavelength spectroscopies measure attenuance, the depletion of light reaching the detector by both scattering or absorption of photons by an analyte, but over time absorbance has become the accepted, if incorrect, term. Here, we use attenuance and absorbance in their formal senses.’

- Something is off with the labeling in Figure 1 and in the text. While the figure shows only panels A to D, panels up to H appear in the legend and in the text.

3.13. We apologise. See 0.2 for the explanation.

- How was Figure 1E or H (the purified supercomplex) created? Was this also measured in the 5 ml chamber, explaining the rasterized spectrum? What concentration was used?

3.14. The rasterisation was due to the low resolution image, which is now fixed. See lines 669-673 in the methods for how the spectrum was measured.

- mOD: Does this stand for OD divided by 1000?

3.15. Yes.

- One should write out the actual concentrations of CCCP, BDQ, and nigericin in the text. This is important given the fact that the outcomes are different. Using terms like low and higher seems not appropriate.

3.16. Good idea, this has now been done.

- Figure 2A: What is the reason for the temporary drop in OCR at 1.5 μ M BDQ?

3.17. This will be the effect of the solvent bedaquiline was dissolved in (ethanol), which has extra O_2 in.

- Figure 3G: What is the signal around 580 nm? It has not been addressed.

3.18. As the spectra is relative to baseline everything is in ‘difference’ mode and therefore it is possible for there to be components of the spectrum that are negative (i.e. absorb less in one state than in another). See the below, extremely carefully collected, reference haem difference spectra for bacterial bc_1 . Depending on these exact changes, these negative components will be more or less visible but are completely normal for haem difference spectra.

Difference spectra from Shinkarev et al, BBA, 2006.

- I didn't find an explanation in the text, why 30 and 120 nM BDQ are used for Msm and Mtb.

3.19 We chose concentrations that were low but still gave a robust effect. As we did not extensively explore the low concentrations we would not want to be drawn on whether even less could be added. Certainly, we add far less bedaquiline than previous studies. Lamprecht et al saw little effect at 3xMIC (162 nM) and had to work at 1.62 μ M. Hards et al 2015 used 3.6 μ M. Most likely this discrepancy comes because we did not include detergent in our wash step, so there was no hydrophobic phase for the bedaquiline to dissolve into. We originally preferred this approach because we were worried about off-target effects, although we see no evidence for them when working at 10 μ M.

- Line 162–163: Why is a more dense culture required for the Mtb compared to the Msm?

3.20 Mtb consumes O₂ slowly so we were able to put more in and get a better signal:noise ratio. While this matters little in this study, in the future it may be useful.

- Line 171–172: It is acting like an oligomycin-like inhibitor, except for the lack of OCR decrease. OCR decrease is a cornerstone of oligomycin activity. I know it is mentioned in the next paragraph, but standing alone, the statement is not correct.

3.21. Agreed. It now reads: ‘This increase is consistent with bedaquiline inhibiting ATP synthase, blocking the major conduit of protons back into cells and increasing PMF.’ (line 261 & 262)

- Line 178–179: The statement is only partially correct. Next to electroneutral proton release to the periplasm, there is an electrogenic uptake of protons to the catalytic site in bd oxidases which contributes to the $\Delta\psi$ and thus the pmf.

3.22. Thank you for this. Revisiting the cryo-EM papers, we have realised you are correct, we should have picked this up. The sentence now reads (line 269-272):

‘Unlike the *aa*₃-type oxidase in the supercomplex (Cta), CydAB lacks vectorial proton pumping activity, it only builds the PMF as the ‘chemical’ protons for the reduction of O₂ to H₂O most likely come from the cytoplasm, and electrons from the oxidation of MQH₂ move from the periplasm.^{19,20}’

- Line 512: 10 mM DCCD is high. Which concentration was used?

3.23. We are very sorry but have realised this number is wrong. It should read 10 μ M. This has now been corrected.

Reviewer #2 (Remarks to the Author):

The authors satisfactorily addressed all my questions.

Supportive- no response required. Thank you

Reviewer #3 (Remarks to the Author):

Harrison et al have done a very thorough job addressing all the comments of the different reviewers. The revised version of the manuscript is now significantly longer and addresses critical points. The authors acknowledge that not all data can be perfectly explained but they provide possible explanations and testable ideas. And I hope they will test some of these explanations in the future. The methods applied here combining visible light spectroscopy and OCR measurements is a pioneering work and reflects well the complexity of bioenergetics in general or cellular redox balance in particular. I am still puzzled by some of the results they get (e.g. non-reduced a type hemes, different OCR simulations, nigericin results), but I acknowledge that they have done these experiments as thoroughly as possible and they deserve publications in these form. As pointed out by the authors, future studies, including such with purified membranes or proteins will test their ideas on how to activate bd oxidases within seconds or if yet another redox balance has to be considered.

Please find below some small minor comments that occurred to be during reading.

P3, line 85. According to the internet, saprotrophic should be used instead of saprophytic, if the organism in question is not a plant.

Fixed, thank you

P7, line 217. Instead of writing generic Figure 2, the exact panels should be specified. Here 2A, D. If other unprecise instances exist, they should also be corrected.

Thank you, this has been carried out and we have checked the other figures

P7, line 221-223. The interpretation with the off-target effects at 25 μ M CCCP is possibly correct, but the amount of data shown here might not substantially support this interpretation.

Given that this is a tentative statement ('presumably') we are comfortable with our wording.

P7, line 233. I don't understand how nigericin builds a $\Delta\psi$, if doing electroneutral K/H (1:1) exchange. This is different behavior than valinomycin, which follows Nernst-potential behavior.

We acknowledge that the precise mechanism remains unclear, but we make no claim to understand it fully. What is important, however, is that it is demonstrably distinct from the action of bedaquiline.

P7, 235-6. The nigericin behavior is a riddle to me. A large drop in OCR means a higher backpressure, resulting in higher cytochrome occupancy, but the opposite is seen. Then after a few minutes the whole recovers. But the authors do a good job describing their observations.

Thank you

Page 10, line 326: great should be greatly

Thank you, fixed.